# The relationship between spatial configuration and functional connectivity of brain regions

**Janine Diane Bijsterbosch[1]\*, Mark W Woolrich[2], Matthew F Glasser[3,4], Emma C Robinson[5], Christian F Beckmann[6,7], David C Van Essen[3], Samuel J Harrison[1†], Stephen M Smith[1†]**

[1]Centre for Functional MRI of the Brain, Wellcome Centre for Integrative Neuroimaging, Nuffield Department of Clinical Neurosciences, University of Oxford, Oxford, United Kingdom; [2]Centre for Human Brain Activity, Wellcome Centre for Integrative Neuroimaging, Department of Psychiatry, University of Oxford, Oxford, United Kingdom; [3]Department of Neuroscience, Washington University Medical School, Missouri, United States; [4]St. Luke's Hospital, Missouri, United States; [5]Department of Biomedical Engineering, School of Biomedical Engineering and Imaging Sciences, King's College London, London, United Kingdom; [6]Donders Institute, Radboud University Medical Centre, Nijmegen, Netherlands; [7]Department of Cognitive Neurosciences, Radboud University Medical Centre, Nijmegan, Netherlands

**\*For correspondence:**
Janine.Bijsterbosch@ndcn.ox.ac.uk

[†]These authors contributed equally to this work

**Reviewing editor:** Chris Honey,

**Abstract** Brain connectivity is often considered in terms of the communication between functionally distinct brain regions. Many studies have investigated the extent to which patterns of coupling strength between multiple neural populations relates to behaviour. For example, studies have used 'functional connectivity fingerprints' to characterise individuals' brain activity. Here, we investigate the extent to which the exact spatial arrangement of cortical regions interacts with measures of brain connectivity. We find that the shape and exact location of brain regions interact strongly with the modelling of brain connectivity, and present evidence that the spatial arrangement of functional regions is strongly predictive of non-imaging measures of behaviour and lifestyle. We believe that, in many cases, cross-subject variations in the spatial configuration of functional brain regions are being interpreted as changes in functional connectivity. Therefore, a better understanding of these effects is important when interpreting the relationship between functional imaging data and cognitive traits.
DOI: https://doi.org/10.7554/eLife.32992.001

## Introduction

The organisation of the human brain into large-scale functional networks has been investigated extensively over the past two decades using resting state functional magnetic resonance imaging (rfMRI). Spontaneous fluctuations in distinct brain regions (as measured with rfMRI) show temporal correlations with each other, revealing complex patterns of functional connectivity (FC) (*Biswal et al., 1995*; *Friston, 1994*, *2011*). Extensive connectivity between cortical areas and with subcortical brain regions has long been considered a core feature of brain anatomy and function (*Crick and Jones, 1993*), and dysfunctional coupling is associated with a variety of neurological and psychiatric disorders including schizophrenia, depression, and Alzheimer's disease (*Castellanos et al., 2013*). Given the great potential neuroscientific and clinical value of rfMRI, it is

**eLife digest** People differ a lot from one another in terms of their personality, behaviour and lifestyle. This individuality is attributed to the different regions in the brain, and the strength of communication between them. The connectivity pattern between these areas is thought to be as unique as a fingerprint. If the connections are weak or disrupted it can play a role in conditions such as schizophrenia, depression or Alzheimer's disease. It is thought that the strength of the connection depends on how strongly the nerve cells in these regions communicate. But are these individual differences solely caused by different strengths of connection, or could other factors contribute to them?

Now, Bijsterbosch et al. found that the size, shape and exact position of the brain regions was also strongly linked to the different behaviours of individuals. The study used brain scans, behavioural tests and questionnaires from a large database about lifestyle choices and demographics, to analyse the relationship between the different brain features of healthy individuals. The results showed that the variations in the brain regions were linked to many behavioural factors including intelligence, life satisfaction, drug use and aggression problems. Moreover, Bijsterbosch et al. showed that the existing methods for estimating the strength of connection between brain regions could reveal more about the spatial layout of these regions than the actual connection strength between them. This suggests that new approaches are needed to properly evaluate the strength of the connections.

Some psychiatric and neurological diseases may be associated with changes in size and position of the different regions in the brain. In future, the findings of this study could be applied to individuals affected by such conditions, to see if the location of a region could be used as a diagnostic indicator.

DOI: https://doi.org/10.7554/eLife.32992.002

important to determine which aspects of rfMRI data most sensitively and interpretably reflect trait variability across subjects. At a neural level, potential sources of meaningful cross-subject variability include: (i) the strength of the functional coupling (i.e. interactions) between two different neural populations ('*coupling*'), and (ii) the spatial configuration and organisation of functional regions ('*topography*'). In this study, we aim to identify how these key aspects of rfMRI data influence derived measures of functional connectivity and how they relate to interesting trait variability in behaviour and lifestyle across individuals. Our findings reveal variability in the spatial topography of functional regions across subjects, and suggest that this variability is the primary driver of cross-subject trait variability in correlation-based FC measures obtained via group-level rfMRI parcellation approaches. These results have important implications for future rfMRI research, and for the interpretation of FC findings.

A commonly applied approach used to derive FC measures from rfMRI data is to parcellate the brain into a set of functional regions ('nodes'), and estimate the temporal correlations between pairs of node timeseries ('edges') to build a network matrix (*Smith et al., 2013b*). This approach has previously been likened to a fingerprint, enabling the unique identification of individuals, and the prediction of behavioural traits such as intelligence (*Finn et al., 2015*; *Passingham et al., 2002*). Of particular interest is the ability of network matrices to explain cross-subject variability in behaviour and performance on psychometric tests. To this end, Cross Correlation Analysis (CCA) was previously adopted to link a 'positive-negative' axis of behaviour to network matrices in data from the Human Connectome Project (*Smith et al., 2015*). CCA allows the comparison of a set of variables obtained from rfMRI (such as network matrices of edges) to a set of behavioural variables by estimating independent linear transformations for the two sets of variables such that they are maximally correlated. Here, we replicated this previous work in a larger subject sample (almost double the number of individuals), and adopt CCA to determine which key aspect of rfMRI data is uniquely associated with behaviour.

Parcellation methods that can be used to estimate network matrices include the use of anatomical, functional, and multi-modal atlases (*Glasser et al., 2016*; *Tzourio-Mazoyer et al., 2002*; *Yeo et al., 2011*), with functional parcellations often being data driven via techniques such as

clustering and independent component analysis (ICA) (*Beckmann et al., 2005*; *Craddock et al., 2012*). Data-driven approaches such as ICA have been used to identify consistent large-scale resting state networks (*Damoiseaux et al., 2006*) and to characterise FC abnormalities in a variety of mental disorders (*Littow et al., 2015*; *Pannekoek et al., 2015*). Any given parcellation is typically defined at the group level, and hence additional steps are required to map a group-level parcellation onto individual subjects' data (that has undergone registration to a common space), in order to obtain subject-specific parcel timeseries and associated connectivity edge estimates. Timeseries derived from hard (binary, non-overlapping) parcellations are often obtained using a simple masking approach (i.e. extracting the averaged BOLD timeseries across all voxels or vertices in a node), whereas ICA parcellations (partially overlapping, soft parcellations that contain continuous weights) are mapped onto single-subject data using dual regression analysis or back projection (*Calhoun et al., 2001*; *Filippini et al., 2009*). The first stage of a dual regression approach involves multiple spatial regression of group ICA maps into each preprocessed individual dataset to obtain subject-specific timeseries; the second stage is a multiple temporal regression of these stage one timeseries into the same preprocessed dataset to obtain subject-specific spatial maps. Note, dual regression is, to some extent, expected to underestimate subject-specific spatial variability because it involves post-hoc regressions of a group-level set of spatial maps, which are unlikely to be an accurate model for the data of individual subjects. Indeed, previous work has shown that, in the presence of spatial variability or inaccurate intersubject alignment, these common methods for mapping group parcellations onto individuals do not recover accurate subject-specific functional regions, and this can severely impact the accuracy of estimated timecourses and derived FC edges (*Allen et al., 2012*; *Smith et al., 2011*).

More recently, several studies have developed more thorough characterisations of the patterns of spatial variability in network topography across subjects (i.e. spatial shape, size and position of functional regions) (*Glasser et al., 2016*; *Gordon et al., 2017a*, *2017b*; *Laumann et al., 2015*; *Swaroop Guntupalli and Haxby, 2017*; *Wang et al., 2015*). For example, Glasser et al. showed that the subject-specific spatial topology of area 55b in relation to the frontal and premotor eye fields substantially diverged from the group average in 11% of subjects (*Glasser et al., 2016*). In addition, the size of all cortical areas, including large ones like V1, varies by twofold or more across individuals (*Amunts et al., 2000*; *Glasser et al., 2016*). This extensive presence of spatial variability across individuals highlights the need for analysis methods that are adaptive and better able to accurately capture functional regions in individual subjects. Another approach that aims to achieve a more accurate subject-specific description of this spatial variability is PROFUMO, which simultaneously estimates subject and group probabilistic functional mode (PFM) maps and network matrices (instead of separate parcellation and mapping steps). Specifically, PROFUMO is a matrix factorisation model that decomposes data into estimates of subject-specific spatial maps, time courses, and amplitudes using a variational Bayesian approach with both spatial and temporal priors that seek to optimise for both spatial map sparsity and temporal dynamics consistent with haemodynamically regularised neural activity (*Harrison et al., 2015*). PROFUMO adopts a hierarchical approach by iteratively optimising subject and group estimates (instead of first estimating group components using group ICA and separately mapping these onto subjects using dual regression), and is therefore expected to more accurately capture subject-specific spatial variability than does dual regression. Other approaches are available to obtain group and subject parcellations in one step, for example using a groupwise normalised cut spectral clustering approach (*Shen et al., 2013*). In the present study, we show that the spatial variability across subjects captured in PFMs is strongly associated with behaviour.

Conceptually, network edges are commonly thought of as reflecting coupling strength between spatially separated neuronal populations. However, as discussed above, edge estimates are highly sensitive to spatial misalignments across individuals. Additionally, correlation-based edge estimates are influenced by the amplitudes of localised spontaneous rfMRI fluctuations (*Duff et al., 2018*), which have been shown to capture trait variability across subjects, and state variability within an individual over time (*Bijsterbosch et al., 2017*). These findings demonstrate the sensitivity of edge-strength estimates to many different types of subject variability, and highlight the need to identify which aspects of FC tap most directly into behaviourally relevant population-level variability. Here, we investigate the complex relationships between different features of an rfMRI dataset and also the associations with variability across individuals in terms of their performance on behavioural tests,

their lifestyle choices, and demographic information. Using data from the Human Connectome Project (HCP), we provide evidence for systematic differences in the spatial organisation of functional regions. We then use simulations that manipulate aspects of the data such that, for example, only cross-subject spatial variability is present in the data (i.e. by fixing edge strength to be the group average for each individual) to investigate whether these differences reflect meaningful cross-subject information and drive edge estimates for several common FC approaches.

## Results

### Cross-subject information in fMRI-derived measures

To determine whether a given rfMRI-derived FC measure contains meaningful cross-subject information rather than random variability, we adopted an approach that makes use of the extensive set of behavioural, demographic, and lifestyle data acquired in the HCP. Our first analysis aims to determine which measures obtained from rfMRI and task data most strongly relate to interesting behavioural variability across individuals. Using Canonical Correlation Analysis (CCA), we extracted population modes of cross-subject covariation that represent maximum correlations between combinations of variables in the subject behavioural measures and in the fMRI-derived measures, uncovering multivariate relationships between brain and behaviour. For example, previous work has used CCA on HCP data to identify a mode of population covariation that linked a positive-negative axis of behavioural variables to patterns of FC edge strength (*Smith et al., 2015*). A specific pattern of connectivity, primarily between 'task-negative' (default mode) regions (*Raichle et al., 2001*), was found to be linked to scores on positive factors such as life satisfaction and intelligence, and inversely associated with scores on negative factors such as drug use.

CCA works by finding a linear combination of behavioural measures (V) that is maximally correlated with a linear combination of rfMRI-derived measures (U). CCA scores for each subject are obtained for the behavioural and fMRI-derived measures (V and U), which represent the subject's position along the population continuum for the latent CCA variable(s). The key result of a CCA analysis for each mode of covariation is the correlation between U and V, denoted $r_{UV}$, which describes the strength of the multivariate brain-behaviour relationship. Given that CCA explicitly optimises $r_{UV}$, it is essential to perform permutation testing in order to test the significance of the CCA result. To determine which behavioural measures contribute strongly to the CCA result, V is subsequently regressed into original non-imaging variables (*Figure 1B*; although interpretation of these results is complicated by behaviour-behaviour correlations). Additionally, U is used to visualise variation at both the population extremes (see *Figure 2* below and *Figure 2—figure supplements 2–7*), and across the full population continuum (Supplementary video files).

We applied a separate CCA analysis for each of the various fMRI-derived measures (including spatial, network matrix, and amplitude measures). The results (*Figure 1* and *Supplementary file 1a and b*) reveal that highly similar associations with behaviour and life factors occur across a wide range of different fMRI-derived measures. Correlating the behavioural subject weights (V) across the different CCA instances in *Figure 1* shows that a similar behavioural mode is obtained from the independent instances of CCA (particularly for those CCAs that have a high $r_{U-V}$ and low $P_{U-V}$; *Figure 1—figure supplement 1*). Mapping these subject weights onto behaviour through correlation reveals consistent positive associations with, for example, fluid intelligence, life satisfaction, and delayed discounting, and consistent negative correlations with use of tobacco, alcohol and cannabis. All behavioural correlations with mean correlation $r > |0.25|$ (chosen for visualisation purposes) are shown in *Figure 1B*. The results show that spatial features such as PFM subject spatial maps and subject task contrast maps are strongly associated with behaviour. Overall, these findings reveal that a large variety of fMRI measures have similarly strong associations with behaviour.

Direct comparison between the results in *Figure 1—figure supplement 1a*) and the HCP_MMP1.0 parcellation (e.g. the 360-region 'Glasser parcellation' [*Glasser et al., 2016*]) and against associated fractional surface area (in native space as a ratio to total surface area, for each of the 360 parcels in the HCP_MMP1.0 parcellation) is challenging due to the large difference in the number of subjects (n = 819 for *Figure 1* and n = 441 for HCP_MMP1.0). Therefore, we have included an analysis on all PFM metrics in a reduced number of subjects (the same n = 441 subjects) in order to facilitate direct comparison between these two recent parcellation approaches that both

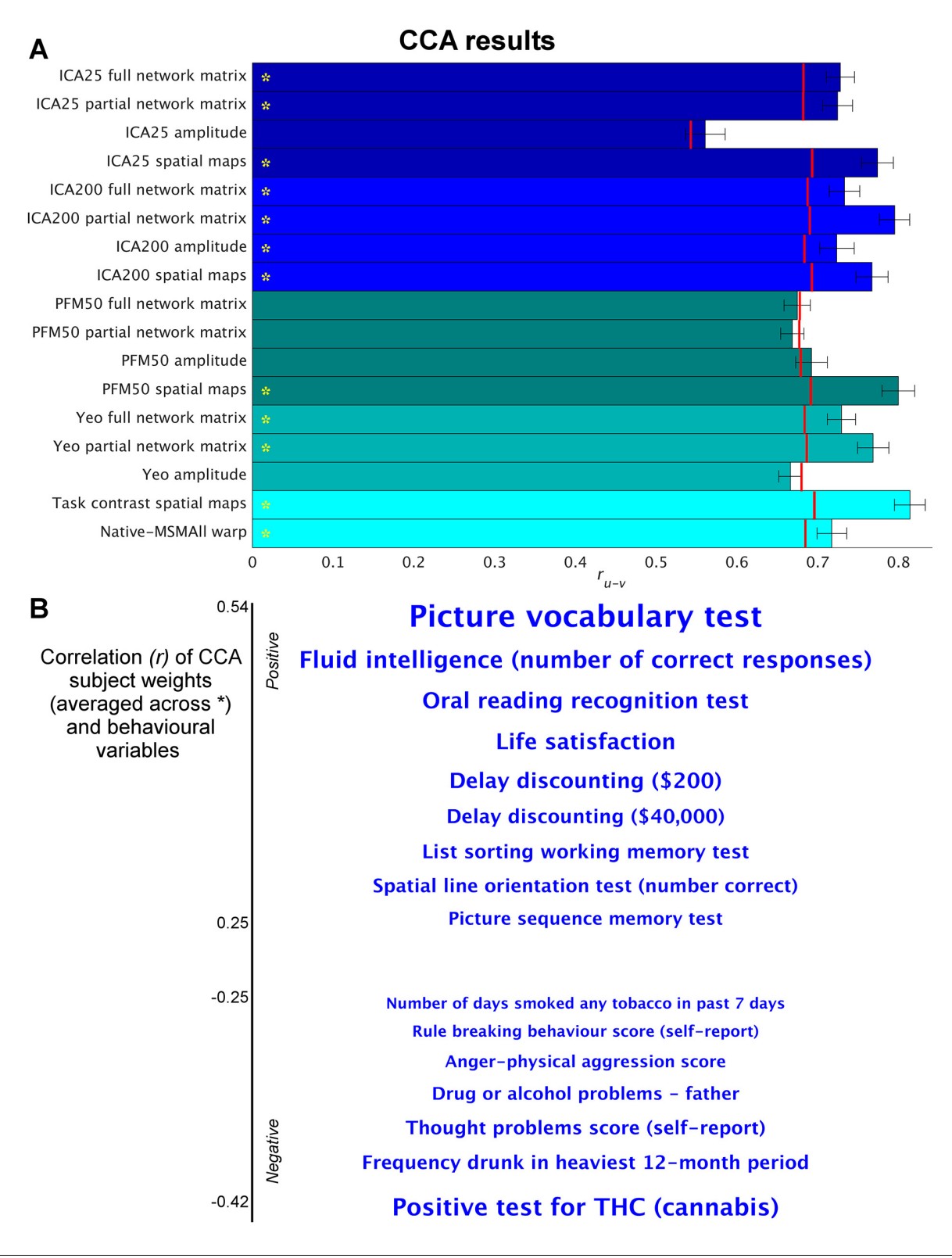

**Figure 1.** Highly similar associations between behaviour and the brain occur across 17 distinct measures derived from fMRI. (A) Comparison of strength of CCA result for network matrices, spatial maps and amplitudes (node timeseries standard deviation) derived from several distinct group-average spatial parcellations/decompositions: ICA decompositions at two scales of detail (dimensionalities of 25 and 200, with 'ICA200 partial network matrix' corresponding to the measures used previously [*Smith et al., 2015*]); a PROFUMO decomposition (PFM; dimensionality 50); an atlas-based hard

*Figure 1 continued on next page*

*Figure 1 continued*

parcellation (108 parcels [*Yeo et al., 2011*]), task contrast spatial maps (86 contrasts, 47 unique), and warp field from native space to MSMAll alignment. Each bar reports a separate CCA analysis (first CCA mode shown), performed against behaviour/life-factors. A similar mode of variation is found across most of the parcellation methods and different fMRI measures. $r_{UV}$ is the strength of the canonical correlation between imaging and non-imaging measures. Error bars indicate confidence intervals (2.5–97.5%) estimated using surrogate data (generated with the same correlation structure), and red lines reflect the p<0.002 significant threshold compared with a null distribution obtained with permutation testing (i.e. family-wise-error corrected across all CCA components and Bonferroni corrected across a total of 25 CCAs performed, see *Supplementary file 1a and b* for the full set of results). CCA estimates the highest possible $r_{uv}$ given the dataset; therefore, the null distribution for low-dimensional brain data (e.g. ICA 25 amplitude) is expected to be lower than for high-dimensional brain data. (B) Set of non-imaging variables that correlate most strongly with the CCA mode (averaged subject weights V across results marked with * in A; i.e. p=0.00001) with behavioural variables. Position against the y-axis and font size indicate strength of correlation.

DOI: https://doi.org/10.7554/eLife.32992.003

The following source data and figure supplement are available for figure 1:

**Source data 1.** Source data for *Figure 1*.
DOI: https://doi.org/10.7554/eLife.32992.004
**Figure supplement 1.** Similarity of behavioural subject weights from a range of separate CCA analyses between MRI-derived measures and behavioural measures.
DOI: https://doi.org/10.7554/eLife.32992.005

aim to achieve accurate detection of subject-specific spatial boundaries (*Supplementary file 1b*). These results show that spatial features from a variety of sources (surface area, multimodal parcellation and PFMs) are strongly associated with measures of behaviour and lifestyle. Also note that network matrices obtained by the HCP_MMP1.0 parcellation are more predictive of behaviour than are PFM network matrices.

For correlation-based parcellated FC estimates (network edges), a common assumption is that functional coupling is primarily reflected in the edges. In theory, true network coupling information can be manifested along a continuum ranging from spatial maps to network matrices. On one extreme, coupling information is purely contained in spatial maps, as is the case when performing temporal ICA (where the temporal correlation matrix is by definition the identity matrix [*Smith et al., 2012*]). On the other extreme, coupling information can be fully contained in network matrices as is often assumed to be the case when using an individualised hard parcellation (however, coupling can only be represented fully in edge estimates if all subjects are perfectly functionally aligned to the parcellation, and if the node timeseries amplitudes do not contain useful cross-subject information). It is likely that the dimensionality of the decomposition may influence this; for example, for a low-dimensional decomposition (into a small number of large-scale networks), much cross-subject variation in functional coupling is likely to occur between sub-nodes of the networks, which is therefore more likely to be represented in the spatial maps, whereas in a higher dimensionality decomposition this information is more likely to be represented in the network matrix. However, the results in *Figure 1* show that this CCA mode of population covariation is significantly present in both spatial maps and network matrices for both low- and high-dimensional decompositions (ICA 25 and 200). Therefore, the potential role of dimensionality is not sufficient to explain the common information present in spatial maps, timeseries amplitudes, and network matrices.

The presence of this behaviourally meaningful spatial variability is somewhat surprising, because these data were aligned using a Multimodal Surface Matching (MSM) approach (*Robinson et al., 2014*; *2018*), driven by both structural and functional cortical features (including myelin maps and resting state network maps). MSM has been shown to achieve very good functional alignment compared with other methods, and particularly compared with volumetric alignment approaches or surface-based approaches that use cortical folding patterns rather than areal features (*Coalson et al., 2018*). However, residual cross-subject spatial variability is still present in the HCP data after the registration to a common surface atlas space (in part due to the constrained parameterisation of MSM and in part because weighted regression subject maps used to drive MSM may not fully capture all spatial variability). In line with this, approaches which are expected to better identify residual subject spatial variability (specifically, PFM spatial maps and subject task contrast maps in *Figure 1*) show strong correspondence between spatial variability and behaviour/life-factor measures.

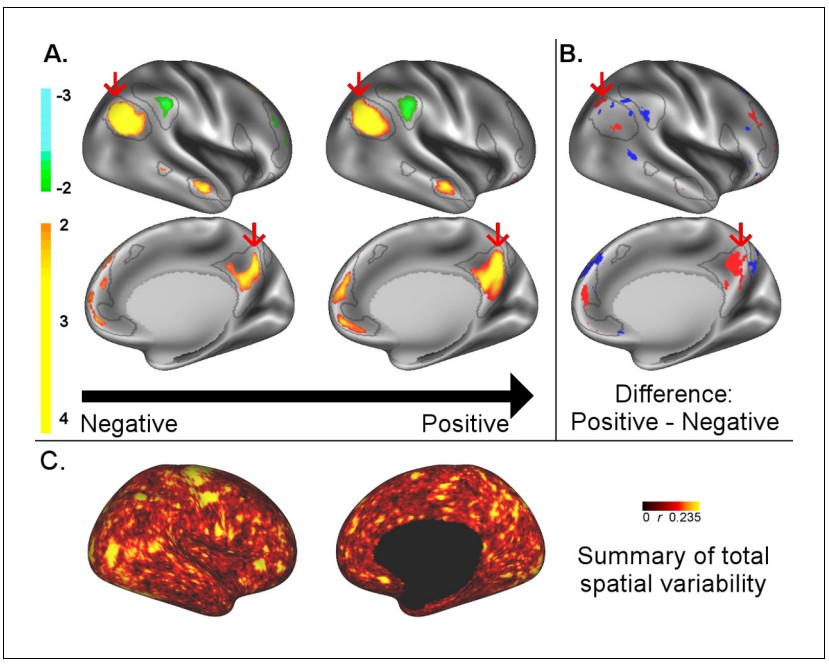

**Figure 2.** A: representative maps of the two extreme ends (identified based on the low and high extremes along a linearly spaced vector that spans the full range of subject CCA scores) of the CCA mode of population covariation continuum are shown for the default mode network (DMN, the PFM mode that contributed most strongly to the CCA mode of population covariation). The top row shows that the inferior parietal node of the DMN differs in shape and extends into the intraparietal sulcus in subjects who score high on the positive-negative CCA mode (right), compared with subjects who score lower (left). The bottom row shows that medial prefrontal and posterior cingulate/precuneus regions of the DMN differ in size and shape as a function of the CCA positive-negative mode. The representative maps at both extremes are thresholded at ±2 (arbitrary units specific to the PFM algorithm) for visualisation purposes (the differences are not affected by the thresholding; for unthresholded video-versions of these maps, please see the Supplementary video files. The grey contours are identical on the left and right to aid visual comparison and are based on the group-average maps (thresholded at 0.75). Spatial changes of all PFM modes can be seen in the Supplementary video files and in *Figure 2—figure supplements 2–7*. B: difference maps (positive - negative; thresholded at ±1) are shown to aid comparison. C: A summary of topographic variability across all PFM modes, showing PFM correlations with CCA subject weights (at each grayordinate the maximum absolute r across all PFMs is displayed). An extended version of C is available in *Figure 2—figure supplement 7*. Data of *Figure 2* available at: https://balsa.wustl.edu/8lVx.

DOI: https://doi.org/10.7554/eLife.32992.006

The following figure supplements are available for figure 2:

**Figure supplement 1.** Representative maps of the two extreme ends of the positive-negative continuum for five PFMs.

DOI: https://doi.org/10.7554/eLife.32992.007

**Figure supplement 2.** Representative maps of the two extreme ends of the positive-negative continuum for five PFMs.

DOI: https://doi.org/10.7554/eLife.32992.008

**Figure supplement 3.** Representative maps of the two extreme ends of the positive-negative continuum for five PFMs.

DOI: https://doi.org/10.7554/eLife.32992.009

**Figure supplement 4.** Representative maps of the two extreme ends of the positive-negative continuum for five PFMs.

DOI: https://doi.org/10.7554/eLife.32992.010

**Figure supplement 5.** Representative maps of the two extreme ends of the positive-negative continuum for five PFMs.

DOI: https://doi.org/10.7554/eLife.32992.011

**Figure supplement 6.** Representative maps of the two extreme ends of the positive-negative continuum for five PFMs.

DOI: https://doi.org/10.7554/eLife.32992.012

*Figure 2 continued on next page*

*Figure 2 continued*

**Figure supplement 7.** Comparison of the cortical representation of associations with behaviour across fractional area, HCP_MMP1.0 individual subject parcellation and PFM spatial maps.

DOI: https://doi.org/10.7554/eLife.32992.013

To better understand what spatial features represent behaviourally relevant cross-subject information, we visually explored what aspects of the PFM spatial maps contributed to the CCA result in *Figure 1* by calculating representative maps at extremes of the CCA mode of population covariation (based on CCA subject scores). While the PFM maps are estimated using the full set of cortical and subcortical grayordinates, we focus on cortical findings because these contribute most strongly to the CCA results. The results reveal complex changes in spatial topography (*Figure 2*, *Figure 2—figure supplements 2–7*, and *Videos 1–9*. For example, comparing left versus right panels shows the right inferior parietal node of the DMN extending farther into the intraparietal sulcus (in the vicinity of area IP1 [*Choi et al., 2006*; *Glasser et al., 2016*]) in subjects who score higher on the behavioural positive-negative mode of covariation. Qualitative inspection of *Figure 2—figure supplements 2–7* suggests that many of the difference maps show notable bilateral symmetry.

## Spatiotemporal simulations demonstrating potential sources of variability in edges

*Figure 1* showed that functionally-relevant cross-subject variability is represented in a variety of different measures derived from both resting state and task fMRI. These widespread similarities in correlations with behaviour across a range of measures invite the question of whether the same type of trait variability is meaningfully and interpretably reflected in a wide range of rfMRI measures, or whether (for example) estimates of network matrices may instead primarily reflect trait variability in spatial topography or amplitude (and not coupling strength). Therefore, we wanted to determine to what extent correlation-based FC measures derived from rfMRI can be influenced by specific aspects of the rfMRI data such as true topography and true coupling. To this end, we generated simulated datasets based on the original PFM subjects and/or group spatial maps and timeseries. By holding either the individual (simulated) subjects' spatial maps or the network matrices fixed to the group average we eliminated specific forms of underlying subject variability from the simulated data (*Figure 3*). Note, we used PFMs in order to generate simulated data because the PROFUMO model separately estimates spatial maps, network matrices and amplitudes, thereby allowing each aspect to be fixed to the group average prior to generating simulated data using the outer product (as described in detail in *Equation (1)*, and in the section on 'Creating simulated data' in the Material

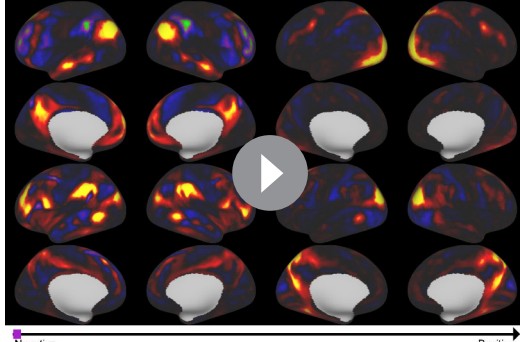

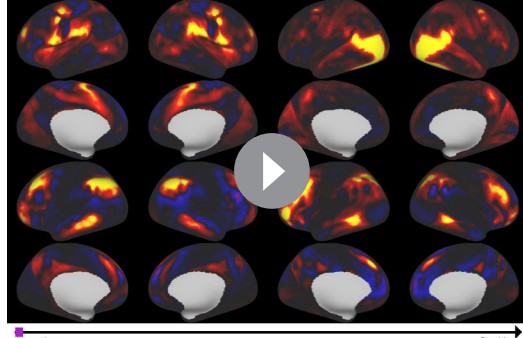

**Video 1.** Unthresholded maps are shown for the 4 PFMs that contribute most strongly to the CCA result (14, 45, 35, 33; corresponding stills in *Figure 2* and *Figure 2—figure supplement 1*). Each video shows five frames representing the continuum from negative to positive CCA results.

DOI: https://doi.org/10.7554/eLife.32992.014

**Video 2.** Unthresholded maps are shown for the next 4 PFMs that contribute most strongly to the CCA result (following earlier video files; 22, 1, 8, 48; corresponding stills in *Figure 2—figure supplements 1,2*). Each video shows five frames representing the continuum from negative to positive CCA results.

DOI: https://doi.org/10.7554/eLife.32992.015

and methods). Previous simulation results have shown that PROFUMO is able to accurately estimate spatial maps and network matrices in the presence of cross-subject variability in spatial topography, relative strength of subregions, and between-mode connectivity (*Harrison et al., 2015*). The aim of the simulation analyses was to determine which features in the rfMRI data are likely to be most strongly reflected in network matrices estimated from rfMRI data. We assess this in terms of the amount of variability across subjects that can be explained, as this is the most relevant application in biomarker studies and in neuroimaging research more generally.

Timeseries were extracted from both the simulated and original datasets, and network matrices were estimated. Each simulated dataset was assessed using three metrics: (i) comparing subject-specific simulated and original network matrices ($Z_{network\ matrix}$ in *Table 1*), (ii) comparing cross-subject variability in the simulated and original network matrices ($R_{correlation}$ in *Table 1*), and (iii) determining how much of the cross-subject variability in simulated and original network matrices is behaviourally informative using CCA (see *Table 1* legend).

The results (*Table 1* and *Supplementary file 1c and d*) show that, when the subject-varying aspects of the simulations were exclusively driven by spatial changes across subjects (with the predefined network matrix and amplitudes being identical for all subjects), up to 62% (i.e. square of $R_{correlation}$ = 0.79 from *Supplementary file 1d* 'maps only') of the cross-subject variance present in the network matrices obtained from the original data was regenerated. Hence, this finding reveals that very similar network matrices can be obtained for any individual subject even if the only aspect of the rfMRI that is varying across subjects is the topographic information in PFM spatial maps. In addition, the variance that can be explained by spatial maps is behaviourally relevant; the CCA results were similarly strong (typically having the same permutation-based p-values) from simulated network matrices driven purely by spatial changes, compared with those obtained from the original dataset.

The influence of amplitudes on FC estimates was relatively minor (less than 2.5% of variance was explained by amplitude in all our simulations; i.e. square of $R_{correlation}$ = 0.15 from *Table 1* 'amplitudes only'), although, when amplitudes were combined with spatial maps feeding into the simulations, the amplitudes did in most cases result in an increase in original network matrix regeneration.

Given the complex information present in PFM spatial maps, the effect of spatial information on network matrices can result from cross-subject variability in: (i) network size, (ii) relative strength of regions within a given network, or (iii) size and spatial location of functional regions. We performed two further tests to distinguish these influences by thresholding and binarising the subject-specific spatial maps used to create the simulated data. Maps were either thresholded using a fixed threshold (removing the influence of relative strength), or (separately) using a percentile threshold (removing the influence of relative strength and size, as the total number of grayordinates in binarised PFM maps is fixed across subjects and PFMs). The role of subject-varying spatial maps in driving the resulting estimated network matrices remains strong when highly simplified binarised maps are used to drive the simulations (*Supplementary file 1e*), further supporting our interpretation that the results are largely driven by the shape of the functional regions (i.e. variability in the location and shape of functional regions across subjects), rather than by size or local strength.

## Unique contribution of topography versus coupling

The results presented above show that a large proportion of the variance in estimated network matrices is also represented in spatial topography. This suggests either that cross-subject information is represented in both the coupling strength between neural populations and in the 'true' underlying spatial topography, or that edge estimates obtained from rfMRI data primarily reflect cross-subject spatial variability (which indirectly drives edge estimates through the influence of spatial misalignment on timeseries extraction, particularly when group parcellations are mapped onto individual subjects in the case of imperfect alignment). To test these hypotheses further, we investigated the unique information contained in spatial maps and network matrices using a set of 15 ICA basis maps derived from HCP task contrast maps (*Figure 4A*). These basis maps can be thought of as the spatial building blocks that can be linearly combined to create activation patterns for any specific HCP task contrast, and can be considered here to be another functional parcellation.

The advantage of using basis maps derived from task data is that the tasks essentially act as functional localisers that allow for the precise localisation of task-related functional regions within an individual; results at a single-subject level are not influenced in any way, including spatially, by the group results, as they are derived via the standard task-paradigm analysis (i.e. which relies solely on

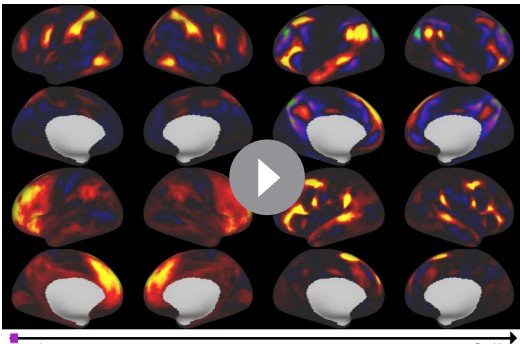

**Video 3.** Unthresholded maps are shown for the next 4 PFMs that contribute most strongly to the CCA result (following earlier video files; 4, 26, 15, 6; corresponding stills in *Figure 2—figure supplements 2,3*). Each video shows five frames representing the continuum from negative to positive CCA results.
DOI: https://doi.org/10.7554/eLife.32992.016

temporal information, and is not influenced by the group-level maps). The equivalence between group- and subject-level contrasts (i.e. the inherent assumption in any group-level analysis, namely that the group '2BK-0BK' contrast map directly relates to any subject-level '2BK-0BK' contrast) means that any combination of group-level contrasts is equally valid as a combination at the subject-level, but with the advantage that the resulting subject maps will be faithful to the precise location of functional regions that the subject-specific contrast maps capture. Hence, subject-based task basis maps are the most accurate description of subject-specific locations of functional regions, at least with respect to those regions identifiable from the range of tasks used.

To investigate the implications of these task-localised maps on typical rfMRI analyses, either group-based task basis maps or subject-based task basis maps were entered into a dual regression analysis against subjects' resting-state fMRI data to obtain network matrices (from dual regression stage one timeseries) and rfMRI-based spatial maps (from dual regression stage 2) for each subject (*Figure 4B*). Subsequently, CCA was performed to determine how well each of the group-based and subject-task-based rfMRI maps and network matrices was able to predict behavioural variability. Furthermore, a 'partial CCA' was performed to characterise the unique variance that task rfMRI maps carry over and above network matrices, and vice versa. Here, we regressed any variance explained by network matrices out of the spatial maps prior to running the 'partial CCA' to determine the unique information contained in spatial maps (and vice versa, i.e., regressed any variance explained by spatial maps out of network matrices before running the 'partial CCA').

The results from the CCAs against behavioural measures show that subject-specific spatial maps (derived from either subject- or group-based task-fMRI maps) capture more behavioural information than network matrices (and continue to reach significance in the partial CCA), consistent with the PFM spatial results presented in *Figure 1*. The full CCA result is marginally stronger ($\Delta$ruv=0.005, p=0.46) for group-task-based rfMRI spatial maps compared with subject-task-based rfMRI spatial

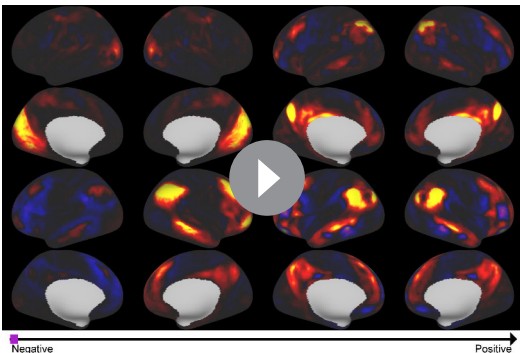

**Video 4.** Unthresholded maps are shown for the next 4 PFMs that contribute most strongly to the CCA result (following earlier video files; 40, 12, 50, 46; corresponding stills in *Figure 2—figure supplement 3*). Each video shows five frames representing the continuum from negative to positive CCA results.
DOI: https://doi.org/10.7554/eLife.32992.017

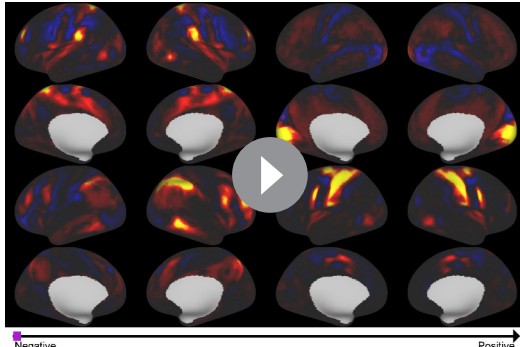

**Video 5.** Unthresholded maps are shown for the next 4 PFMs that contribute most strongly to the CCA result (following earlier video files; 18, 9, 43, 2; corresponding stills in *Figure 2—figure supplement 4*). Each video shows five frames representing the continuum from negative to positive CCA results.
DOI: https://doi.org/10.7554/eLife.32992.018

maps. None of the partial CCA results for network matrices reach significance, suggesting that network matrices do not contain any unique trait-level information that is not also captured by spatial variability. Note that dual regression maps derived from both group-task-based and subject-task-based templates capture unique subject-specific spatial variation in the partial CCA results, consistent with significant CCA results for ICA25 and ICA200 in *Figure 1*. Importantly, subject-task-based rfMRI network matrices explain the behavioural data considerably less well than group-based task-rfMRI network matrices (difference: p=0.0005 for full network matrices), confirming that spatial information is a significant factor in estimated network matrices. Hence, subject spatial variability is more uniquely represented in the spatial information for subject-task-based estimates, and therefore does not filter through into the network matrices (marked "§").

Taken together, these results show that, while network matrices obtained from dual regression against group-level maps do contain behaviourally relevant cross-subject information, this can be almost completely explained by variability in spatial topographical features across subjects (to the extent that we can detect it). Hence, dual regression network matrices (obtained from multiple regression against group spatial maps) apparently contain little unique cross-subject information regarding coupling strength that is not also reflected in spatial topographical organisation. However, it is possible that network matrices obtained using parcellation methods and timeseries extraction approaches that are better able to capture subject-specific spatial variability (such as the HCP_MMP1.0 parcellation) do contain unique cross-subject information; further research is needed to test this possibility. Additionally, network matrices may contain unique state-level information relevant to ongoing behaviour (e.g. in a task paradigm).

## Discussion

Here, we have identified a key aspect of rfMRI data that directly reflects interesting variability in behaviour and lifestyle across individuals. Our results indicate that spatial variation in the topography of functional regions across individuals is strongly associated with behaviour (*Figure 1*). In addition, network matrices (as estimated with masking or dual regression against group-level hard or soft parcellations) reflect little or no unique cross-subject information that is not also captured by spatial topographical variability (*Figure 4* and *Figure 4—figure supplement 1*). This unexpected finding implies that the common interpretation of FC as representing cross-subject (trait) variability in the coupling strength of interactions between neural populations may not be a valid inference (although within-subject state-dependent changes in coupling may still be reflected in FC measures). Specifically, we show that up to 62%

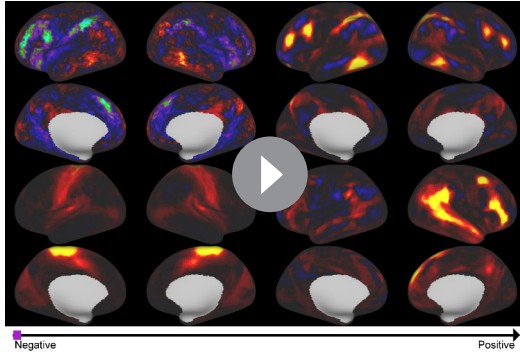

**Video 6.** Unthresholded maps are shown for the next 4 PFMs that contribute most strongly to the CCA result (following earlier video files; 29, 11, 37, 24; corresponding stills in *Figure 2—figure supplements 4,5*, map 29 is missing from stills because results fall below the still threshold). Each video shows five frames representing the continuum from negative to positive CCA results.
DOI: https://doi.org/10.7554/eLife.32992.019

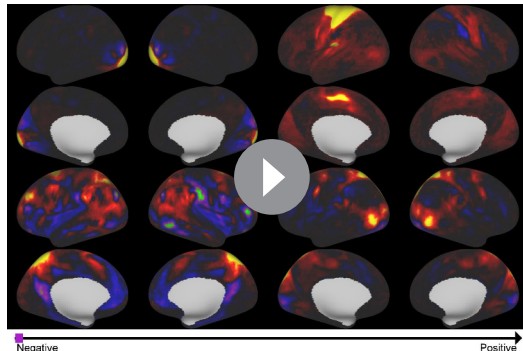

**Video 7.** Unthresholded maps are shown for the next 4 PFMs that contribute most strongly to the CCA result (following earlier video files; 10, 38, 20, 39; corresponding stills in *Figure 2—figure supplements 5,6*). Each video shows five frames representing the continuum from negative to positive CCA results.
DOI: https://doi.org/10.7554/eLife.32992.020

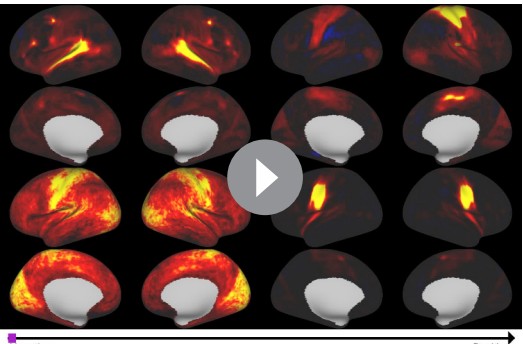 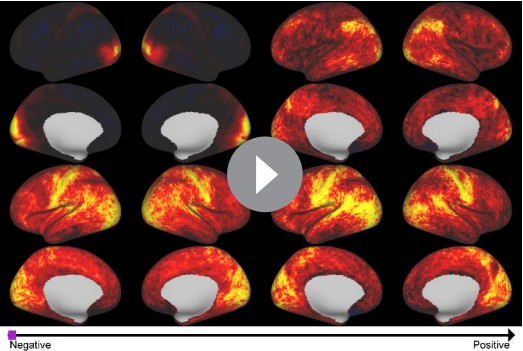

**Video 8.** Unthresholded maps are shown for the next 4 PFMs that contribute most strongly to the CCA result (following earlier video files; 49, 7, 19, 30; corresponding stills in *Figure 2—figure supplement 6*, map 19 is missing from stills because results fall below the still threshold). Each video shows five frames representing the continuum from negative to positive CCA results.
DOI: https://doi.org/10.7554/eLife.32992.021

**Video 9.** Unthresholded maps are shown for the next 4 PFMs that contribute most strongly to the CCA result (following earlier video files; 17, 3, 42, 23; corresponding stills in *Figure 2—figure supplement 4*, maps 3, 42, 23 are missing from stills because results fall below the still threshold). Each video shows five frames representing the continuum from negative to positive CCA results.
DOI: https://doi.org/10.7554/eLife.32992.022

of the variance in rfMRI-derived network matrices (a measure commonly taken as a proxy for coupling) can be explained purely by spatial variability. These findings have important implications for the interpretation of FC and may contribute to a deeper mechanistic understanding of the role of intrinsic FC in cognition and disease (*Mill et al., 2017*).

Our findings are consistent with previous research that has highlighted the presence of structured cross-subject spatial variance in both functional and anatomical networks (*Glasser et al., 2016*; *Gordon et al., 2017a*; *Noble et al., 2015*; *Sabuncu et al., 2016*; *Tong et al., 2017*; *Xu et al., 2016*). Furthermore, recent work has shown that resting state spatial maps can be used to predict task activation maps from individual subjects very accurately (*Tavor et al., 2016*), and that interdigitated and highly variable subnetworks can be identified within individuals (*Braga and Buckner, 2017*). Therefore, the presence of behaviourally relevant cross-subject variance in maps of functional (co-) activation in itself is not surprising. However, the fact that these variations in spatial topographical features capture a more direct and unique representation of subject variability than temporal correlations between regions defined by group parcellation approaches (coupling), was unexpected. The implication of this finding is that the cross-subject information represented in commonly adopted 'connectivity fingerprints' largely reflects spatial variability in the location of functional regions across individuals, rather than variability in coupling strength (at least for methods that directly map group-level parcellations onto individual data). Specifically, our partial CCA results (*Figure 4*) show that network matrices (as often estimated) contain little unique trait-level cross-subject information that is not also reflected in the spatial topographical organisation of functional regions.

How the functional organisation of the brain is conceptualised and operationally defined is of direct relevance to the interpretation of these findings. Some hard parcellation models of the human cortex (such as the Gordon and Yeo parcellations [*Gordon et al., 2016*; *Yeo et al., 2011*]) aim to fully represent connectivity information in the edges (i.e. correlations between node timeseries). Thus, hard parcellations of this type assume piecewise constant connectivity within any one parcel (i.e. each parcel is assumed to be homogeneous in function, with no state- or trait-dependent within-parcel variability in functional organisation). In contrast, the HCP_MMP1.0 multimodal parcellation presumes within-area uniformity of one or more major features, but overtly recognises within-area heterogeneity in other features, including connectivity, most notably for distinct body part representations ('sub-areas') of the somatomotor complex. Soft parcellation models (such as PROFUMO [*Harrison et al., 2015*]) allow for the presence of multiple modes of (potentially overlapping) functional organisation. Therefore, PFMs represent connectivity information through complex interactions between amplitude and shape in the spatial maps, and through network matrices. Our findings show that both the PROFUMO and the multimodal parcellation models successfully capture

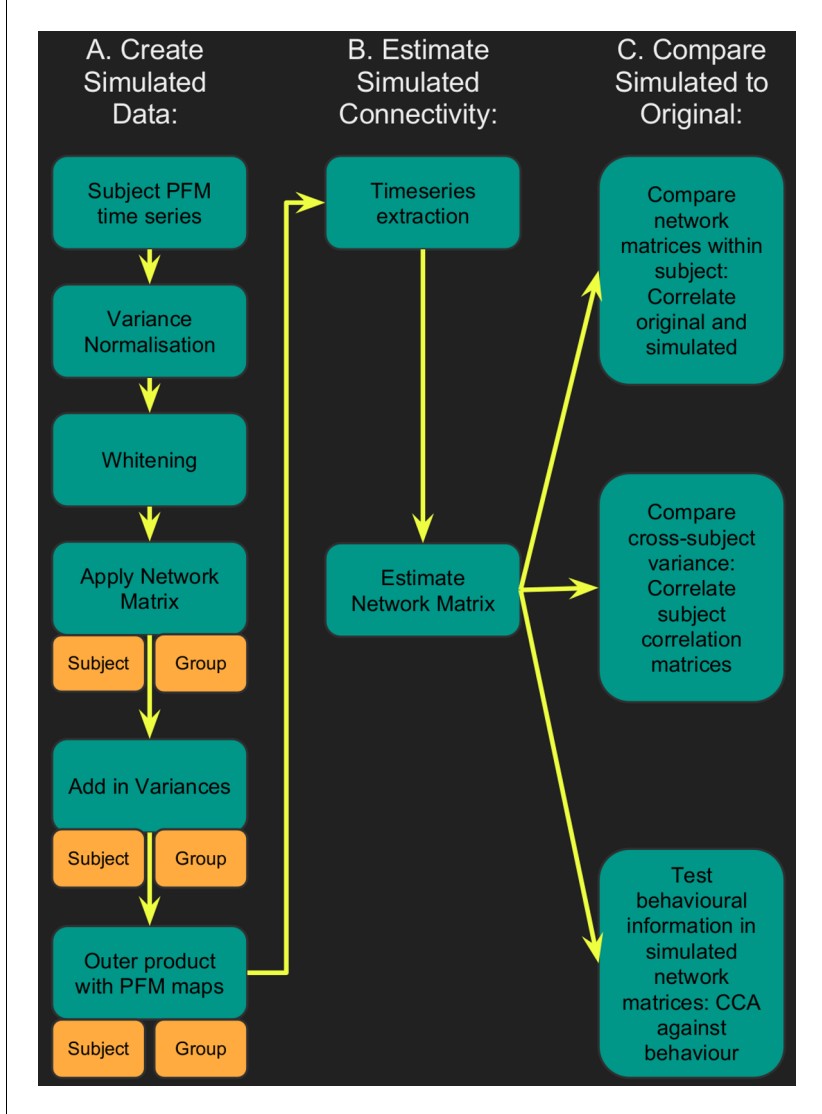

**Figure 3.** Flowchart for spatiotemporal simulations. Simulated data was generated for each subject by setting one or more aspects (from the network matrices, node amplitudes, and spatial maps) to the group average. Timeseries extraction is performed (uinsg either dual regression against original group ICA maps, or masked against a binary parcellation); network matrices are calculated and compared against network matrices estimated from the original data.

DOI: https://doi.org/10.7554/eLife.32992.023

behaviourally relevant cross-subject spatial variability (*Supplementary file 1b*), but that the precise location of where this spatial variability is represented overlaps only modestly between the two approaches (*Figure 2—figure supplement 7*). Given the differences in the key assumptions made by the two models (i.e. binary parcellation versus multiple modes of functional organisation), this is not unexpected. However, it does highlight the need for further research into the optimal representation of (subject-specific) functional organization in the brain.

For most of the results presented in this work, we estimated spatial information using functional data (either resting or task fMRI data). While a comprehensive investigation of related anatomical features is beyond the scope of this work, we did identify significant correlations between fractional surface area size and subject CCA weights (*Figure 2—figure supplement 7*). This result suggests that anatomical variability in the cortical extent of a number of higher level sensory and cognitive brain regions may contribute to the overall findings presented here. Further research into the

**Table 1.** Results from simulated datasets in which one or more of the network matrices, amplitudes and spatial maps are fixed to the group average to remove any subject variability associated with it.

Results in each row were driven by variables in which subject variability was preserved, as indicated with ✓ (variables with '-' were fixed to the group average). Results are shown for within-subject correlations between simulated and original z-transformed network matrices ($Z_{network\ matrix}$), similarities of cross-subject variability represented in simulated and original network matrices ($R_{correlation}$), and for results obtained from the CCA against behaviour (where $r_{U-V}$ is the strength of the canonical correlation between imaging and non-imaging measures, $P_{U-V}$ is the associated (family-wise error corrected) p-value estimated using permutation testing, taking into account family structure, and $r_{U-Uica}$ is the correlation of a CCA mode (subject weights) with the positive-negative mode of population covariation obtained from ICA200 partial network matrices as used in *Smith et al. (2015)*. For brevity, this Table presents results from full correlation network matrices obtained from a dual regression of ICA 200 maps onto the simulated data (because this approach closely matches previously published findings [*Smith et al., 2015*]), results for other parcellations are in *Supplementary file 1c* and for partial correlation network matrices in *Supplementary file 1d*. The results for a wide range of different parcellations show comparable trends (i.e. a large proportion of cross-subject variability is captured purely by spatial maps, as indicated by the highlighted rows), and this main result is also found when using partial network matrices (e.g., for ICA 200, $0.51^2$ = 26% variance explained in partial network matrices was captured by spatial information, and $0.54^2$ = 29% variance explained in full network matrices was captured by spatial information).

| | Simulation driven by true subject variability in: | Network matrix | Amplitude | Spatial map | $Z_{network\ matrix}$ | $R_{correlation}$ | CCA $R_{U-V}$ | Cca $P_{U-V}$ | CCA $R_{U-Uica}$ |
|---|---|---|---|---|---|---|---|---|---|
| ICA D = 200 N = 819 | Nothing | - | - | - | −0.0003 | 0.03 | 0.65 | 0.32017 | 0.11 |
| | Amps and maps | - | ✓ | ✓ | 1.14 | 0.60 | 0.71 | 0.00001 | 0.52 |
| | Connectivity only | ✓ | - | - | 0.47 | 0.65 | 0.69 | 0.00028 | 0.40 |
| | Amplitudes only | - | ✓ | - | 0.22 | 0.15 | 0.69 | 0.00052 | 0.45 |
| | Maps only | - | - | ✓ | 0.78 | 0.54 | 0.72 | 0.00001 | 0.62 |

DOI: https://doi.org/10.7554/eLife.32992.024

relationship between structural features and functional connectivity measures, and their contribution to trait-level subject variability is needed to test this hypothesis.

Our findings are relevant to a wide variety of approaches used to study connectivity. For example, our simulation results (*Table 1* and *Supplementary file 1c and d*) reveal similar results regardless of whether we adopt a dual-regression or a masking approach to obtain timeseries, and the findings also do not differ qualitatively according to whether full or partial correlation is used to estimate network matrices. Therefore, our findings are relevant to any approach that is based on timeseries extracted from functional regions defined at the group-level (including graph theory methods and spectral analyses). The implications of this work may also extend beyond resting-state fMRI. For example, generative models such as dynamic causal modelling (DCM) are increasingly used to stratify patient populations (*Brodersen et al., 2014*), and to achieve predictions for individual patients (*Stephan et al., 2017*). Previous work has shown that including parameters for the position and shape of functional regions in individual subjects into the model improves DCM results and better differentiates between competing models (*Woolrich et al., 2009*). It is currently unknown to what extent cross-subject variability observed with these timeseries-based fMRI metrics reflects true coupling between neural populations, rather than being indirectly driven by spatial variability and misalignment, but given that many of these studies are conducted using alignment methods that perform substantially worse than the MSMAll surface-based alignment used in this study (*Coalson et al., 2018*), this is likely a significant confound for such studies. Going forward, it is important to disambiguate the influence of spatial topography to enable the estimation of fMRI measures that uniquely reflect coupling strength between neural populations.

Significant advances have already been made in recent years in order to tackle the issue of spatial misalignment across individuals. For example, the HCP data used in this work were spatially aligned using the multimodal surface mapping (MSM) technique, which achieves very good functional alignment by using features that are more closely tied to cortical areas (although note that, since the time of the HCP release, refinements to the MSM algorithm and regularisation have resulted in further improvements in the observed functional alignment of HCP data [*Robinson et al., 2014, 2018*]). Therefore, gross misalignment is unlikely to play a role in our results. In fact, some of the

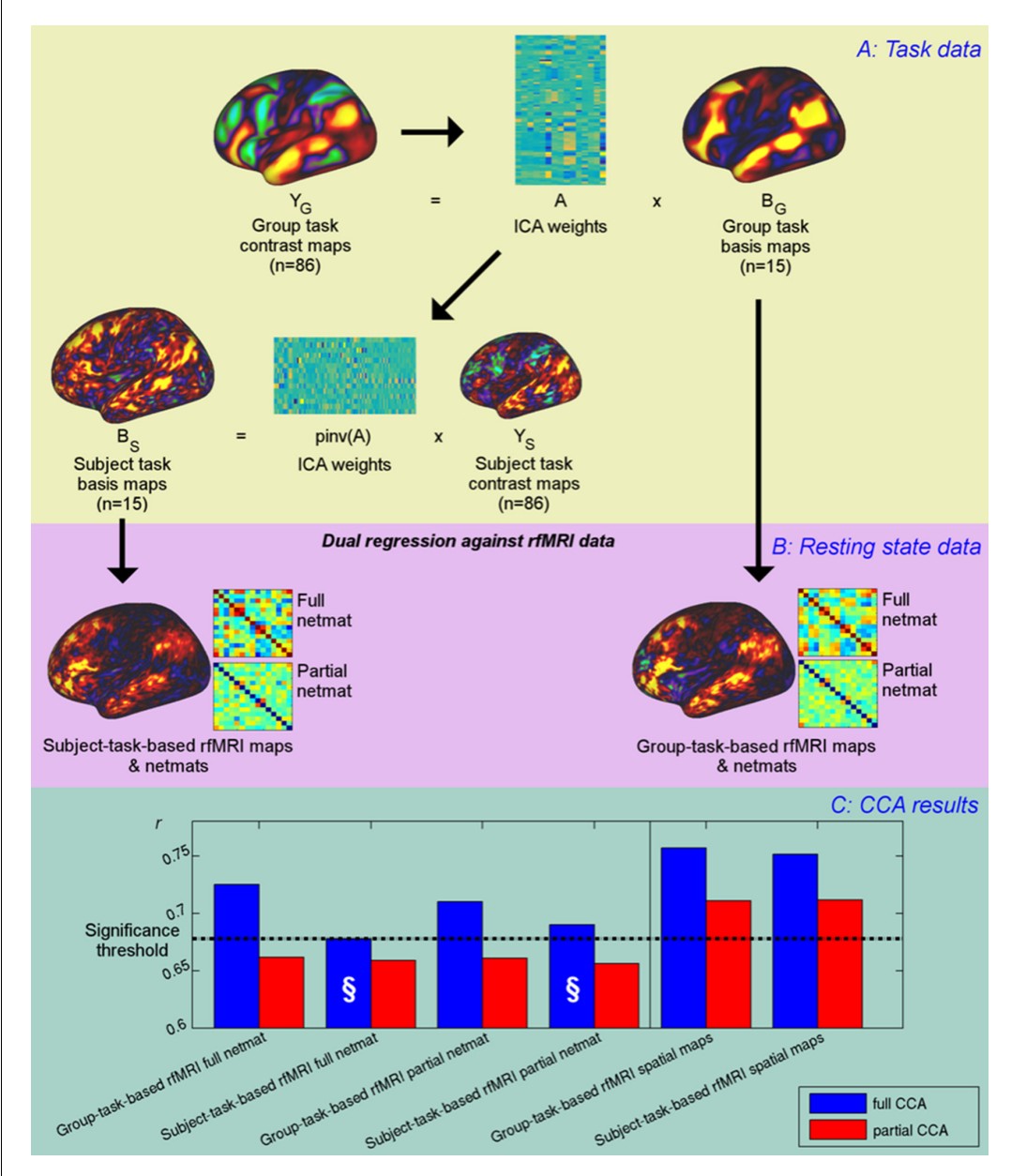

**Figure 4.** Unique contribution of topography versus coupling. (**A**) Task basis maps are extracted from group-averaged task contrasts (n=86, 47 unique) using ICA to ensure correspondence of basis maps across subjects. These maps represent the basic building blocks of any activation pattern, and subject task basis maps (obtained by applying the ICA weights to subject task contrast maps) are not influenced by misalignment problems. (**B**) Dual regression against rfMRI data is performed using either the (potentially misaligned) group task basis maps or the (functionally localised) subject task basis maps. (**C**) CCA results of group-task-based rfMRI maps and network matrices and of subject-task-based rfMRI maps and netmats. The results show rUV (i.e., the correlation between the first U and V obtained from the CCA analysis describing the strength of association between the rfMRI and behavioural measures). The null line (i.e., p=0.05 based on permutation testing) is shown as a dotted line at 0.68; results below this line do not reach significance. The blue bars show the main CCA results using the complete data, and the red bars show partial CCA results computed after regressing out any variance that can be explained by network matrices from the spatial maps and vice versa prior to running the CCA. The results show a general decrease in rUV for all measures when comparing partial to full CCA results. The strongest partial CCA result (red bars on right) are found when using rfMRI spatial maps, and the associated netmats showed the weakest results ("§"). However, the partial CCA results for the spatial maps (i.e., the red bars on the right) still reach significance. All of the partial CCAs also showed lower rU-Uica compared to the full CCAs (not shown here).

DOI: https://doi.org/10.7554/eLife.32992.025

The following source data and figure supplement are available for figure 4:

**Source data 1.** Source data for *Figure 4*.

*Figure 4 continued on next page*

*Figure 4 continued*

DOI: https://doi.org/10.7554/eLife.32992.026

**Figure supplement 1.** Similarities between cross-subject variations estimated from different rfMRI measures.

DOI: https://doi.org/10.7554/eLife.32992.027

behaviourally relevant variability may have been 'corrected' in the MSM pipeline prior to our analyses (indeed, the same positive-negative mode of population covariation is identified when running the CCA on MSM warp fields; and the fractional surface area results in *Supplementary file 1b* and *Figure 2—figure supplement 7* reflect the full variability from native space, and are not affected by the alignment accuracy). Therefore, it is possible that the degree to which spatial information may influence FC estimates varies considerably across studies, depending on the spatial alignment algorithm that was used, and the amount of subject spatial variability this has removed. It is encouraging that significant efforts have recently gone into the methods for more accurately estimating the spatial location of functional parcels in individual subjects in recent years (*Chong et al., 2017*; *Glasser et al., 2016*; *Gordon et al., 2016*; *Hacker et al., 2013*; *Harrison et al., 2015*; *Varoquaux et al., 2011*; *Wang et al., 2015*), and into advanced hyperalignment approaches (*Chen et al., 2015*; *Guntupalli et al., 2016*; *Guntupalli and Haxby, 2017*). The present results highlight the importance of such advances, and call for the continued development, comparison, and validation of such approaches.

In conclusion, we have demonstrated that spatial topography of functional regions are strongly predictive of variation in behaviour and lifestyle factors across individuals, and that timeseries-based methods (as often estimated based on group-level parcellations) contain little unique trait-level information that is not also explained by spatial variability.

# Materials and methods

## Dataset

For this study, we used data from the Human Connectome Project S900 release (820 subjects with fully complete resting-state fMRI data, 452 male, mean age 28.8 ± 3.7 years old) (*Van Essen et al., 2013*). Data were acquired across four runs using multiband echo-planar imaging (MB factor 8, TR = 0.72 s, 2 mm isotropic voxels) (*Moeller et al., 2010*; *Uğurbil et al., 2013*). Data were preprocessed according to the previously published pipeline that includes tools from FSL, Freesurfer, HCP's Connectome Workbench, multimodal spatial alignment driven by myelin maps, resting state network maps, and resting state visuotopic maps ('MSMAll'), resulting in data in the grayordinate coordinate system (*Fischl et al., 1999*; *Glasser et al., 2013*, *2016*; *Jenkinson et al., 2012*; *Marcus et al., 2013*; *Robinson et al., 2014*; *Smith et al., 2013a*). ICA-FIX-cleanup was performed on individual runs to reduce structured noise (*Griffanti et al., 2014*; *Salimi-Khorshidi et al., 2014*). ICA-FIX achieves 99% sensitivity and 99% specificity on HCP data when compared to manual classification by trained raters (*Smith et al., 2013a*). Only subjects with the full 4800 resting state timepoints (4 scans of 1200 TRs each) were included for the analyses performed in this work. A detailed overview of quality assessment in the Human Connectome Project was previously published (*Marcus et al., 2013*).

## Data availability

HCP data are freely available from https://db.humanconnectome.org. The version of MSMAll that is compatible with the approach implemented for the alignment of HCP data can be found here: http://www.doc.ic.ac.uk/~ecr05/MSM_HOCR_v2/ (*Robinson et al., 2017*). Matlab code used in this work can be found here: https://github.com/JanineBijsterbosch/Spatial_netmat (*Bijsterbosch, 2017*; copy archived at https://github.com/elifesciences-publications/Spatial_netmat). Data from many figures in this study is freely available at https://balsa.wustl.edu/study/show/kKM0.

## Inferring functional modes

In order to obtain estimates of the spatial shape and size of functional networks for every subject, we decompose the HCP data into a set of probabilistic functional modes (PFMs) via the PROFUMO

algorithm (*Harrison et al., 2015*). A set of $M$ PFMs describe each subject's data ($G$ grayordinates; $T$ time points; $D_s \in R^{V \times T}$) in terms of a set of subject-specific spatial maps ($P_s \in R^{V \times M}$), amplitudes ($h_s \in R^M$) and timecourses ($A_s \in R^{M \times T}$), all of which are linked via the outer product model:

$$D_s = P_s * diag(h_s) * A_s + \varepsilon \tag{1}$$

These subject-specific decompositions are linked by a set of hierarchical priors. In the spatial domain, the group-level parameters encode the grayordinate-wise means, variances and sparsity of the subject maps, while in the temporal domain, the group-level priors constrain the subject-level network matrices (note that the component amplitudes and hierarchical priors are recent extensions to the PFMs model and were not included in the original PROFUMO paper [*Harrison et al., 2015*]). The PROFUMO framework gives us sensitive estimates of key subject-level parameters, while ensuring that there is direct correspondence between PFMs across subjects.

PROFUMO was run on the rfMRI data from all 820 subjects with a dimensionality of 50 PFMs. Importantly, the signal-subspace of any given subject's dataset can be straightforwardly reconstructed from a set of modes via equation [1], and this can be used to generate the simulated data as described below.

## Canonical correlation analysis (CCA)

For the ICA decompositions, amplitudes were estimated for each subject and component as the temporal standard deviation of the timeseries obtained from stage 1 of a dual regression analysis. Full and regularised partial correlation matrices were also calculated from these timeseries. The Tikhonov regularisation rho used during estimation of the partial correlation matrices was set to 0.01 for the ICA 25, 200 and PFM data (according to previous optimisation results). For high-dimensional parcellations (Yeo and HCP_MMP1.0), the rho was optimised by finding the maximum correlation between subject and group-average (using rho = 0.01) network matrices across a range of rho (0.01:0.5), leading to rho = 0.03 for Yeo and rho = 0.23 for HCP_MMP1.0 results. Lastly, the subject spatial maps obtained from stage 2 of a dual regression analysis were used. Similarly, for the PROFUMO decomposition, the PFM amplitudes, subject spatial maps and timeseries were used. For the HCP_MMP1.0 spatial results, either group-level or subject-specific node parcellations were used (*Hacker et al., 2013*). The subject-specific parcellations contain missing nodes (parcels) in some subjects (*Glasser et al., 2016*). Hence, for partial network matrices, the rows and columns in the covariance matrix were set to the scaled group average prior to inverting the covariance matrix. In the resulting network matrices, the rows and columns relating to missing nodes were set to the group average (for both partial and full network matrices). Before performing CCA, missing nodes were accounted for by estimating the subject-by-subject covariance matrix one element at a time, ignoring any missing nodes for any pair of subjects. The nearest valid positive-definite covariance matrix was subsequently obtained using nearest SPD in Matlab (http://uk.mathworks.com/matlabcentral/fileexchange/42885-nearestspd), prior to performing singular value decomposition as described below.

Each CCA analysis finds a linear combination of behavioural and life-factor measures (V) that is maximally correlated with a linear combination of rfMRI-derived measures (U) (*Hotelling, 1936*): $Y * A = U \sim X * B = V$. Y is the set behavioural measures, and X are the rfMRI-derived measures (i.e. spatial maps, or network matrices, or signal amplitudes), ~indicates that U and V are approximately equal. A and B are optimised such that the correlation between U and V is maximal. Summary measures from CCA include the correlation between (paired columns of) U and V, and the associated p-values (derived from permutation testing over n = 100,000 permutations) for the first one or more CCA modes.

To create the inputs to the CCA, a set of nuisance variables were regressed out of both the behavioural measures and the amplitudes, network matrices and spatial maps, as done in (*Smith et al., 2015*). Subject covariance matrices were subsequently estimated for the amplitudes, network matrices and for all spatial maps (by summing the covariance matrices of individual spatial maps). Then, a singular value decomposition was performed on the subject covariance matrices and the first 100 eigenvectors were entered into the CCA (against 100 eigenvectors obtained from behavioural variables as explained in *Smith et al., 2015*).

In addition to reporting the CCA results for the strength of the canonical correlation between imaging and non-imaging measures and the associated p-value ($r_{U-V}$ and $P_{U-V}$), we also report the correlation between the CCA subject weights and the weights for the ICA 200 partial network matrices ($r_{U-Uica}$). The reason for including this correlation is to facilitate direct comparison to previously published CCA results from HCP data (*Smith et al., 2015*). However, this earlier finding should not be taken as the gold standard CCA result. The $r_{U-Uica}$ correlation we report is the maximum correlation found between the first CCA mode from the ICA 200 partial network matrices, and any of the 100 modes of population covariation obtained for the comparison CCA result (i.e. the maximum correlation may not be with the strongest CCA mode).

Confidence intervals for CCA results in *Table 1* were obtained using surrogate data for both the brain-based CCA input matrix and the behaviour CCA input matrix. To generate the surrogate data, row and column wise correlations of the original CCA input matrices were maintained using a multivariate normal random number generator (mvnrnd.m in Matlab). A total of 1000 instances of surrogate data were used to obtain 2.5–97.5% confidence intervals around $r_{U-V}$.

For visualisation and interpretation purposes, we created videos of the spatial variability along the axis of the behavioural CCA mode of population covariation. For this, we took the U resulting from the CCA between PFM spatial maps and behaviour, and created a linearly spaced vector that spans just over the full range of U (extending beyond the lowest and highest measured subject score by 10% of the full range). As the CCA is linear, it is straightforward to project a set of U values back to form a rank-one reconstruction of the original space, which in this case is a set of spatial maps. This sequence of spatial maps is an approximation to the spatial variability that is encoded along the previously reported positive-negative axis. These are used as the frames for *Videos 1–9* , and for the illustrative examples shown in *Figure 2* and *Figure 2—figure supplements 2–7*.

The two rfMRI parcellation methods included in *Supplementary file 1b* (HCP_MMP1.0 and PFM) explicitly aim to capture cross-subject variability in the spatial location of functional regions. The subject spatial maps estimated by both methods are strongly associated with cross-subject behavioural variability (when matching the sample size $r_{U-V}$ did not significantly differ, and subject weights of the strongest CCA results were moderately correlated $r_{U-U}$ = 0.55). Therefore, it is of interest to compare these results in more detail, to determine whether cross-subject variability is represented similarly for the two approaches. Furthermore, given that fractional surface area (the fraction of cortex occupied by each area in the multimodal HCP_MMP1.0 parcellation) was also strongly predictive of behaviour (*Supplementary file 1b*), we investigated the potential relationship between rfMRI-based PFM weights, multimodally defined cortical areal boundaries (HCP_MMP1.0 parcellation), and structural variation in fractional surface area. To this end, we averaged CCA subject weights obtained from two separate CCA results (PFM spatial maps - behaviour, and HCP_MMP1.0 spatial maps - behaviour). These averaged subject weights were subsequently correlated against fractional surface area, and against subject-specific PFM and HCP_MMP1.0 spatial maps (grayordinate-wise), to investigate which brain regions contribute strongly to the association with behaviour, and to compare these localised effects across methods/modalities.

## Creating simulated data

In order to create simulated datasets for each subject, we took the outer product between PFM spatial maps and timeseries. Compared with data that is completely simulated, this approach has the advantage of keeping many features in the data (such as the types of structured noise that are present, the signal-to-noise ratio, and the autocorrelation structure), while still achieving investigator control of specific aspects of interest. Data from each run (1200 time points) was processed separately through the simulation pipeline, including the following steps:

### Timeseries processing
#### Variance normalisation
Each original PFM subject timecourse was set to unit variance, and the variances were retained.

$$v_s = var(A_s^T); \; B_s = A_s * diag(v_s^{-1/2})$$

## Whitening

The ZCA whitening transform (*Bell and Sejnowski, 1997*) was used to remove any correlations between timeseries: $Z_s = cov(B_s)^{-1/2}$; $C_s = B_s * Z_s$

## Network matrix application

Timeseries were modified such that the induced correlation matched a pre-specified structure.: $D_s = C_s * \alpha$. In the simulations that use a fixed group network matrix, this pre-specified correlation structure was estimated by projecting the S900 group average HCP dense connectome (following Wishart Rolloff) onto the group PFM spatial maps.

## Restore variances

At this stage, the variances of the original timeseries are restored $E_s = D_s * diag(v_s^{1/2})$. This gives a set of simulated timeseries $E_s$ which have all the same properties as the reference timeseries ($A_s$), except for their correlation structure.

## Pseudo-PFM generation

We modify the inferred PFMs by selectively setting some of the parameters to their group averages. For example, if we set $P_s = P_g$, where $P_g$ is the mean over all 820 subject maps, then we can eliminate any spatial variability across subjects. Similarly, we can set the temporal correlations to a fixed group mean using the procedure described above to remove any variability in FC across subjects. In order to remove amplitude variability across subjects, we add in group averaged variances instead of the subject variances. These simulated PFMs are then described by the simulated maps, amplitudes and timeseries, namely $\hat{P}_s$, $\hat{h}_s$ and $\hat{A}_s$.

## Data reconstruction

Finally, the full data can be reconstructed as per [1]: $\hat{D}_s = \hat{P}_s * diag(\hat{h}_s) * \hat{A}_s + \varepsilon$. Spatio-temporally white-noise (with variance matched to the original data) is added to the activity described by the simulated modes to give a dataset that preserves the properties of the original data, but, crucially, one where we have direct control over where in the model subject variability can appear.

Once the simulated data is generated for each run, we extracted timeseries from both the simulated and original data using two different approaches that are commonly adopted in the literature. Dual regression analysis was performed using the group ICA maps that were estimated using the (original) HCP group data, and that are freely available with the S900 data release (www.humanconnectome.org). Two dimensionalities were tested, so for each simulated dataset dual regression was performed against 25 and against 200 group ICA components. The timecourses estimated in stage 1 of the dual regression analysis were used to compute network matrices (*Filippini et al., 2009*; *Nickerson et al., 2017*). Mean timeseries were also extracted from a set of 108 binary regions of interest (ROIs) based on the Yeo parcellation, and from the HCP_MMP1.0 group parcellations and individual subject parcellations (*Glasser et al., 2016*). The 108 Yeo ROIs were obtained from the 17-network parcellation (*Yeo et al., 2011*), by separating each of the 17 networks into individual contiguous regions that had a surface cluster area of at least 20 mm². Timecourses were used to estimate full and regularised partial correlation network matrices using FSLnets (https://fsl.fmrib.ox.ac.uk/fsl/fslwiki/FSLNets). Z-transformation was applied to the network matrices before further comparisons. The network matrices derived from simulated data are compared against network matrices calculated from the original data as described below.

Firstly, we compare the simulated network matrix to the original network matrix for each subject, to determine how similar the measured FC is. For each subject, the node-by-node full or regularised partial network matrix estimated from the simulated data is reshaped into a single column after removing the diagonal and is correlated against the reshaped original estimated network matrix. Prior to reshaping the simulated and original network matrices, the respective group average network matrix (simulated or original) is subtracted from the subject network matrix, so that the subsequent correlation is sensitive to the unique subject variability instead of being driven by the group connectivity patterns. As such, a correlation coefficient between demeaned simulated and original network matrices is estimated for each subject. The Fisher r-to-z transform was applied to these correlations before averaging across subjects. This first test assesses how different a subject is from the

group (and the similarity of this difference between original and simulated network matrices), and therefore does not test for cross-subject variability.

Secondly, the subject-by-subject correlation matrix was estimated from the subject-wise simulated network matrices. Again, this matrix was reshaped into a vector after discarding the diagonal and was correlated against the reshaped subject-by-subject correlation matrix obtained from the original network matrices. The aim of this test was to directly compare the cross-subject variability present in the simulated and original data, which is very important given that variability across subjects is typically of primary interest in FC research. Hence, this analysis aims to compare the cross-subject variability in original or simulated network matrices, as opposed to comparing the similarity of original and simulated network matrices within an individual subject (as is the case for the preceding approach).

The last test of the simulated network matrices was to perform a CCA against the set of behavioural and life-factor measures (*Smith et al., 2015*). A CCA was performed on the simulated network matrices against the subject behavioural measures as described below. To assess the CCA results, we report the correlation between U and V (for the first, strongest mode of population covariation), the associated permuted p-value (n = 100,000 permutations, respecting family structure), and the maximum correlation between any of the simulated U and the first U obtained when using the original ICA 200 dimensionality partial network matrices describing the positive-negative mode of covariation (*Smith et al., 2015*).

## Simulations with further spatial map modulations

The PFM subject spatial maps contain a relatively complex set of information. This may include relative differences in amplitude in different brain regions that are part of the same mode, which effectively reflect connectivity rather than spatial shape and size. In order to exclude these potential connectivity-related aspects of the spatial maps and isolate the role of spatial shape, we simplified the spatial maps for some of the simulations presented. For this, the spatial maps were thresholded at a very liberal threshold of 1 (arbitrary units specific to the PFM algorithm) and binarised. The sign was retained such that grayordinates in the subject PFM maps with values > 1 were set to one and grayordinates with values <-1 were set to −1 and all others to zero. A liberal threshold was purposefully used as we wanted to retain extended (broad, low) shape information, and just remove any information encoded in the (relative) grayordinate amplitudes. Using a fixed threshold across subjects retains cross-subject variability in the size of networks. To further remove this source of information and focus purely on the shape of networks, we applied a percentile threshold such that the size of networks is fixed across subjects (grayordinates >95th percentile set to one and grayordinates <5th percentile set to −1, leading to each individual PFM map having the same size of 4564 1s and 4564 −1s across all subjects). The results of simulations where the maps were modulated in this way prior to calculating the simulation's space-time outer product are presented in *Supplementary file 1e*, including results for which the maps were both thresholded and binarised, percentile thresholded and binarised, and also results for maps that were thresholded (at 1) but not binarised.

## Comparing cross-subject similarities between different types of imaging measures

Given that variability between subjects is of primary interest in rfMRI research, this analysis aimed to directly compare the cross-subject variability present in a range of measures obtained from the original data. Between-subject correlation matrices were calculated from network matrices (ICA25, ICA200 and PFM50), from PFM amplitudes and from spatial maps (ICA25 and ICA200 dual regression stage two spatial maps, and PFM50 spatial maps). These subject by subject correlation matrices were reshaped after discarding the diagonal, and full and partial correlations were calculated between the subject correlation matrices (*Figure 4—figure supplement 1*).

## Unique contribution of topography versus coupling

To obtain a basis set of spatial maps based on task contrast data, we performed a spatial ICA (with a dimensionality of 15) on the concatenated group-averaged task contrast maps (a total of 86 maps, 47 of which are unique). The ICA dimensionality was determined based on the proportion variance

explained in the PCA data reduction step (99.0% for d = 15). Spatial ICA was performed on the group-average task contrasts maps to avoid the correspondence problem that would arise if ICA were applied separately to individual subject task contrast maps. This resulted in a set of ICA weights (15*86), which describe the contribution of each task contrast map to each extracted ICA component. The outer product of these weights with either the group-averaged contrast maps or the corresponding subject-specific contrast maps was used to obtain maps to drive subsequent dual regression analysis. Dual regression analysis (driven by either group-averaged or subject-specific task basis maps after normalising the maximum of each subject and component map to 1) was run against subject resting state data to obtain timeseries and maps. CCA against behaviour was performed separately on the resulting network matrices and spatial maps as described above. Additionally, partial CCA was performed to determine the unique information contained in network matrices and in spatial maps. For this, any variance explained by network matrices was regressed out of the spatial maps and vice versa (i.e. was 'partialled out'), before running the 'partial CCA'. Specifically, the 100 eigenvectors used as the input matrix to the CCA (as explained above and following [*Smith et al., 2015*]) for partial network matrices were regressed out of the 100 eigenvectors for the spatial maps before running CCA, or conversely the 100 eigenvectors for spatial maps were regressed out of the 100 eigenvectors for the network matrices before running CCA.

## Acknowledgements

Data were provided by the Human Connectome Project, WU-Minn Consortium (Principal Investigators: David Van Essen and Kamil Ugurbil; 1U54MH091657) funded by the 16 NIH Institutes and Centers that support the NIH Blueprint for Neuroscience Research; and by the McDonnell Center for Systems Neuroscience at Washington University. CFB acknowledges support from The Netherlands Organization for Scientific Research (NWO, grant no 864.12.003). We are grateful for funding from the Wellcome Trust (grants 098369/Z/12/Z and 091509/Z/10/Z). The Wellcome Centre for Integrative Neuroimaging is supported by core funding from the Wellcome Trust (203139/Z/16/Z).

## Additional information

### Competing interests

David C Van Essen: Reviewing editor, *eLife*. The other authors declare that no competing interests exist.

### Funding

| Funder | Grant reference number | Author |
| --- | --- | --- |
| National Institutes of Health | 1U54MH091657 | David C Van Essen |
| Wellcome Trust | 098369/Z/12/Z | Stephen M Smith |
| Nederlandse Organisatie voor Wetenschappelijk Onderzoek | 864-12-003 | Christian F Beckmann |
| Wellcome Trust | 091509/Z/10/Z | Stephen M Smith |
| Wellcome Trust | 203139/Z/16/Z | Stephen M Smith |

The funders had no role in study design, data collection and interpretation, or the decision to submit the work for publication.

### Author contributions

Janine Diane Bijsterbosch, Conceptualization, Software, Formal analysis, Investigation, Visualization, Methodology, Writing—original draft; Mark W Woolrich, Investigation, Methodology, Writing—review and editing; Matthew F Glasser, Data curation, Investigation, Methodology, Writing—review and editing; Emma C Robinson, Christian F Beckmann, Software, Methodology, Writing—review and editing; David C Van Essen, Data curation, Funding acquisition, Investigation, Methodology, Writing—review and editing; Samuel J Harrison, Conceptualization, Software, Formal analysis,

Investigation, Methodology, Writing—review and editing; Stephen M Smith, Conceptualization, Supervision, Funding acquisition, Investigation, Methodology, Writing—review and editing

### Author ORCIDs
Janine Diane Bijsterbosch (iD) http://orcid.org/0000-0002-1385-9178
David C Van Essen (iD) http://orcid.org/0000-0001-7044-4721
Samuel J Harrison (iD) https://orcid.org/0000-0002-5886-2389

### Ethics
Human subjects: HCP data were acquired using protocols approved by the Washington University institutional review board. Informed consent was obtained from subjects. Anonymised data are publicly available from ConnectomeDB (db.humanconnectome.org; Hodge et al., 2016). Certain parts of the dataset used in this study, such as the age of the subjects, are available subject to restricted data usage terms, requiring researchers to ensure that the anonymity of subjects is protected (Van Essen et al., 2013).

### Decision letter and Author response
Decision letter https://doi.org/10.7554/eLife.32992.036
Author response https://doi.org/10.7554/eLife.32992.037

## Additional files
### Supplementary files
• Supplementary file 1. (**a**) Highly similar associations between behaviour and the brain can be found across a wide range of different measures derived from fMRI. We included a set of network matrices, spatial maps and amplitudes (node timeseries standard deviation) derived from several distinct group-average spatial parcellations/decompositions: from ICA decompositions at two scales of detail (dimensionalities of 25 and 200); a PROFUMO decomposition (PFM; dimensionality 50); an atlas-based hard parcellation (108 parcels [*Yeo et al., 2011*]); task contrast spatial maps (86 contrasts); and MSM warp fields from native space to MSMAll aligned data (from estimate_metric_distortion; https://github.com/ecr05/MSM_HOCR_macOSX/blob/master/src/MSM/estimate_metric_distortion.cc). Each row reports a separate CCA analysis, performed against behaviour/life-factors. A very similar mode of variation is found across most of the parcellation methods and different fMRI measures. $r_{U-V}$ is the strength of the canonical correlation between imaging and non-imaging measures (confidence intervals estimated using surrogate data), $P_{U-V}$ is the associated (family-wise error corrected) p-value estimated using permutation testing, taking into account family structure, and $r_{U-V}$ CI is the 2.5–97.5% confidence interval estimated using surrogate data. $r_{U-Uica}$ is the correlation of a CCA mode (subject weights) with the positive-negative mode of population covariation obtained from ICA200 partial network matrices as used in *Smith et al. (2015)*, and is therefore defined to be one in the row containing the results from that CCA. The $r_{U-Uica}$ result was included because it shows whether different metrics are associated with similar or distinct behavioural modes of population covariation (one may expect different rfMRI measures to be associated with distinct aspects of behaviour). The final column contains the total number of CCA modes with $P_{U-V}$ <0.05 (results in other columns correspond to the most significant CCA mode, except for $r_{U-Uica}$, which relates to the maximum correlation across all CCA modes). (**b**) The $r_{U-V}$ results here are inflated in comparison to the results presented in *Supplementary file 1a* (due to increased overfitting as a result of the parcellation only being available in 441 subjects compared with 819 subjects included for the other CCAs), but the associated $P_{U-V}$ can (to some extent) be used for comparison. Therefore, this Table compares PFM (d = 50), HCP_MMP1.0 (d = 360), and fractional surface area (the fraction of cortex occupied by each area in the multimodal HCP_MMP1.0 parcellation) on the same set of 441 subjects (only considering subjects with a complete set of 4800 resting state timepoints). (**c**) Results from simulated datasets in which one or more of the network matrices, amplitudes and spatial maps are fixed to the group average to remove any subject variability associated with it. Results in each row were driven by variables in which subject variability was present, as indicated with ✓ (variables with - were fixed to the group average). Results are shown for within-subject correlations between simulated and

original z-transformed network matrices ($Z_{network\ matrix}$), across-subject correlations between simulated and original subject correlation matrices ($R_{correlation}$), and for results obtained from the CCA against behaviour. Note that comparable CCA results from the original data can be found in *Supplementary file 1a*. This Table presents results from full correlation network matrices. (**d**) Results from simulated datasets in which one or more of the network matrices, amplitudes and spatial maps are fixed to the group average to remove any subject variability associated with it. Results in each row were driven by variables in which subject variability was present, as indicated with ✓ (variables with - were fixed to the group average). Results are shown for within-subject correlations between simulated and original z-transformed network matrices ($Z_{network\ matrix}$), across-subject correlations between simulated and original subject correlation matrices ($R_{correlation}$), and for results obtained from the CCA against behaviour. This Table presents results from partial correlation network matrices. Note that the results flagged with * are poorly estimated as a result of the low rank of the PFM subject network matrices (containing 50 PFM modes) used to drive these simulations. The reason for this is that the PFM 50-dimensional subject network matrices were added into the data (to keep the simulation pipeline identical). This approximated 50-dimensional network matrix is too low rank to allow accurate estimation of partial connectivity across a much larger number of nodes. The full correlation results in *Supplementary file 1c* are estimable, and support the 25-dimensional ICA results. (**e**) Modulating the subject spatial maps by thresholding and binarizing retains the shape and size aspects, but removes any relative amplitude information from the spatial maps. Binarised % results are binarised after applying a percentile threshold, and therefore only retain shape aspects (while fixing the size). The results reveal that even after thresholding and binarizing the spatial maps, remaining spatial variability strongly drives the cross-subject information present in the resulting network matrices. See earlier Tables for a description of the measures.

DOI: https://doi.org/10.7554/eLife.32992.028

• Transparent reporting form

DOI: https://doi.org/10.7554/eLife.32992.029

## Major datasets

The following dataset was generated:

| Author(s) | Year | Dataset title | Dataset URL | Database, license, and accessibility information |
|---|---|---|---|---|
| Janine Diane Bijsterbosch, Mark W Woolrich, Matthew F Glasser, Emma C Robinson, Christian F Beckmann, David C Van Essen, Samuel J Harrison, Stephen M Smith | 2018 | Study: The relationship between spatial configuration and functional connectivity of brain regions | https://balsa.wustl.edu/study/show/kKM0 | Available on login at the Brain Analysis Library of Spatial maps and Atlases (BALSA) |

The following previously published dataset was used:

| Author(s) | Year | Dataset title | Dataset URL | Database, license, and accessibility information |
|---|---|---|---|---|
| Van Essen D, Ugurbil K | 2017 | Human Connectome Project | https://www.humanconnectome.org/ | Freely available upon agreeing with Open Access Data Use Terms and Restricted Data Use Terms ( https://www. humanconnectome. org/study/hcp-young-adult/document/ quick-reference-open-access-vs-restricted-data). |

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
