## [Decision Letter]

Thank you for submitting your article "The relationship between spatial configuration and functional connectivity of brain regions" for consideration by *eLife*. Your article has been reviewed by four peer reviewers, one of whom is a member of our Board of Reviewing Editors, and the evaluation has been overseen by Sabine Kastner as the Senior Editor. The following individuals involved in review of your submission have agreed to reveal their identity: Jakob Seidlitz (Reviewer #2); Daniel S Margulies (Reviewer #3).

The reviewers have discussed the reviews with one another and the Reviewing Editor has drafted this decision to help you prepare a revised submission.

You can see the specific comments from each reviewer below. During our discussion, the reviewers agreed that this was potentially an important result. We felt that the manuscript would be substantially improved if your team could:

i) Provide more information about the location (regions and subnetworks), scale and direction of the spatial variation that is most consistently observed across participants;

ii) Clarify the logic behind the central claim [that spatial variation accounts for the behaviorally-relevant variation in edge-strength] and show that the analytics are not biased to translate non-spatial sources of variability into spatial sources of variability;

iii) Justify the use of low-dimensional decompositions in some analyses and show whether fine-grained parcellations could ameliorate (or exacerbate) the misattribution effects that are reported;

iv) Provide more information about which specific behavioral metrics covary with the spatial/morphological variation.

We also recommend trying to make the manuscript (especially the Introduction and Results) more readable for those who are familiar with fMRI and functional connectivity methods, but who are not experts in the individual differences sub-field and who have not yet read the Materials and methods in detail.

Reviewer #1:

This manuscript presents a potentially interesting study on the spatial arrangement of functional regions and how they interact, and the relationship of this to non-imaging measures. Overall I think I agree with the premise of the paper (anatomy and function are not independent nor completely separable) and find the result of interest to the brain-behaviour community. However, the paper is really hard to read/follow and its unclear to what extent the authors have proven their point.

The difficulty in following this also makes it very difficult to evaluate the significance of the findings. For example in the first simulation topographic information is added to the HCP data and then it is shown that topographic information accounts for some of the variance. It's not clear how this wouldn't be the case since that is what was added. While the simulation may indeed show that topology accounts for a significant fraction of the variance, this doesn't actually show that this is what is happening in real data. In general, the idea that spatial topology matters, while likely true, is poorly supported by an ad-hoc combination of ICA, parcellation, and PFMs.

Besides the intractable writing there are at least two major concerns:

1) The idea that individualized parcellations can reveal information has been shown previously. Comparing networks derived from group-parcellations with individualized spatial maps is not really a fair comparison. While it makes a nice argument for performing individualized functional parcellations the comparison as set up is not that interesting nor informative. To show that the correlation matrices (networks) do not add any unique information, one would need to show that the differences in parcellations (ICA-component definitions) across subjects can predict behavior better than, or comparable to, networks derived from the individualized functional maps. This is not what was done however.

2) It also appears to be the case that the authors derive the truest subject-specific map (Figure 4) from a group-level ICA. This is mentioned briefly in the Materials and methods section and is necessary to preserve the correspondences in ICA components across subjects. It appears that to maintain correspondence ICA was first performed at the group level and then brought to the individual level. It is not clear that this is a fair approach.

The CCA's in Figure 1 are interesting but somewhat unclear. Independent of the concerns with the simulation expressed above, the interpretation of Figure 1 seems to be that the functional data is confounded by the topographic information but this sort of summary does not show that. I think the authors are implying that the high CCA for topography means that this is the source of the functional CCA but these do not have to be from the same source. The summary data needs to be broken down further to prove that argument.

The only functional (nonICA atlas used) is the Yeo atlas, which was somehow extended from 17 regions up to 109, but this is really a low resolution atlas. It has been shown by many groups that functional atlases of this order (such as AAL) are very poor and there are numerous publicaly available atlases with many more nodes such as the Shen atlas (Shen, 2013, which should be mentioned in the Introduction) 268 nodes, the Glasser atlas (360 nodes), which is discussed but not apparently used. This is particularly important given the comment in subsection “Cross-subject information in fMRI-derived measures” that the dimensionality of the decomposition may influence the results.

While ICA 200, is high relative to the low 25 ICA it is still not high relative to many functional parcellation schemes.

There are many other questions and concerns about various steps in the analysis but these are too numerous to catalog with the manuscript in its current form.

Reviewer #2:

This manuscript directly assesses the relationship between underlying spatial topography and functional connectivity. In general, from my knowledge of the literature, this is the first time that such a direct comparison has been performed in the context of brain-behavior comparisons. Overall, I thought the manuscript was well-written and the analyses well-defined and well-executed, and I believe that this topic (and these findings) are of particular relevance to the wider neuroimaging community, especially given the widespread use of this dataset from the Human Connectome Project. Some notable remarks include the use of high-quality publicly available data from a large cohort, and the reproducibility of the findings across parcellations (and thresholding/binarization). Please find below comments for the authors to consider.

1) What was the justification for the use of a dimensionality of 15 for the creation of the ICA basis maps? It would be good to add this to the Materials and methods section "Unique contribution of topography versus coupling" or within the Results (say in subsection “Unique contribution of topography versus coupling”).

2) How consistent were the loadings of the behavior CCA outputs from each of the comparisons with brain (i.e., Figure 1 in this manuscript)? Is it possible to quantify these relationships across the comparisons in this work? It would be good to evaluate this empirically, with reference to the CCA loadings in Smith et al., 2015 as reference in this manuscript. In addition to this quantitative analysis, the manuscript would benefit from more of a discussion (and potentially a supplemental figure) about the resultant behavior axes derived from the CCA (i.e., similar to Figure 1 from Smith et al., 2015).

3) Given the focus/topic of this manuscript, and as briefly touched on in the discussion (and Figure 1—figure supplement 1) about the relationship between fractional surface area and subject CCA weights, it would be of great interest to the audience of this manuscript to quantify and report the relationship between basic morphological features (for example, cortical thickness, surface area, volume, myelin maps, and/or curvature) and the spatial rfMRI maps/networks. Clearly spatial variability explains a great deal of variance in rfMRI-derived network matrices, but it would be good to compare these maps to classical morphological metrics, which are derivable from these data.

4) I would highly recommend the authors include a section (in the Materials and methods) about the data quality assessment and quality control that was performed, as well as elaborate on the preprocessing procedures. This would be of particular interest to those that would like to try to reproduce these findings or replicate them in an independent dataset.

Reviewer #3:

The study by Bijsterbosch and colleagues addresses the important question of what aspect of functional connectivity varies across individuals. Prior work has largely investigated variance in the strength or amplitude of functional connectivity, however, the current findings demonstrate that much of that variance is better explained by differences in spatial topography. The analyses are conducted on data made available by the Human Connectome Project, and include validation using several network decomposition and parcellation approaches.

The topic of this study is of high relevance to the field, and my primary recommendation to the authors would be to provide more extensive illustration of the topographic variance in the main manuscript. Specifically, Figure 2 is an excellent example of the type of spatial variance underlying individual differences in functional connectivity measures. Further figures depicting the spectrum of spatial variance (based on the maps shown in the supplementary videos, for example) would further help the reader understand the nature of these patterns of interindividual difference. In addition, an additional figure that summarizes the topographic variance across the cortex, along the lines of Figure 1—figure supplement 1, would be a helpful addition to the current manuscript.

Reviewer #4:

Main Issues

It is important to show that the PROFUMO decomposition is not biased to ascribe variance to the spatial mode when that variance arises elsewhere (e.g. from changes in inter-regional correlation). Please include a simulation which explicitly tests this idea (or explicitly describe for the reader the simulations from the prior PROFUMO papers that have demonstrated this fact). It is particularly important to demonstrate this for the case of overlapping generator ROIs. For, example, one could run a simulation on a 40 by 40 voxel grid, with Gaussian sources A, B, C, D located with centers as (10,10), (10,30), (30,10) and (30,30) respectively. The sources are expressed in space as Gaussians with s.d.s of (say) 10 units so that the signals from different sources do overlap. Now, we suppose that the four source signals are generated as a coupled linear system with some coupling matrix. We then ask whether changing the coupling matrix (without changing the spatial profile of the sources) leads PROFUMO to change its inferred spatial modes.For example, in condition 1, the B source is correlated with the C source, and the A source is correlated with the D source. In condition 2, the B source is additionally correlated with the A source. Are the _shapes_ of the sources recovered by Profumo different in Condition 1 and Condition 2? This is important to show, because otherwise the spatial variation inferred by PROFUMO may actually arise from changes in the true coupling matrix. Are the spatial maps the same even under conditions where the true coupling strengths fluctuate around their means over time?

The prediction of behavioral properties using fMRI is a central aspect of this manuscript, and yet there is almost no description of what aspects of human behaviors are being predicted. Which aspects of task performance are being well-captured? In subsection “Cross-subject information in fMRI-derived measures” there is reference to the in "positive-negative mode of covariation" but there is no description of what this means concretely.

Some of the writing in the Results section seems to assume that the readers have already read the Materials and methods, and yet the Materials and methods appear after the results. I suggest providing the readers with a bit more methodological context in the following subsections, to help them understand what was done:

In the Introduction section, when introducing the PROFUMO method, please elaborate a little more on what it does and how it works;

Relatedly, when comparing PROFUMO with dual regression approach, please provide readers with a brief description of the dual regression method, so that they can understand why it may not succeed in accurately recovering subject-specific networks;

Introduction section: "simulations that manipulate aspects of the data" – please be more precise about what is manipulated in the simulations;

Figure 1 – this Figure could be more effective if there was a graphical depiction of how the predictions are made from each of these fMRI-derived measures; as it is now, the reader arrives at this point in the manuscript with little idea of what is meant by any of the row-labels in this Figure;

Results section, paragraph three – once again the reader will have difficulty understanding what is being described because the relevant methods have not yet been laid;

Subsection “Spatiotemporal simulations demonstrating potential sources of variability in edges”; "thereby allowing each aspect to be fixed to the group average prior to generating simulated data using the outer product" – at this point in the manuscripts readers do not have any understanding why an outer product is relevant, or which matrix equation is being referenced;

The authors have quite convincingly shown that spatial variation in fMRI sources across subjects is an important challenge and widespread confound. Please provide some elaboration on your comments in the penultimate paragraph of the Discussion and provide the reader with some specific suggestions on how this problem might be overcome, and in what contexts it is more or less perilous. Is PROFUMO the answer? Is functional co-registration using localizers the answer? Could brain-wide functional hyper-alignment / shared response modeling provide another solution to the spatial variation problem? e.g. Chen et al., 2015.

---

## [Author Response]

You can see the specific comments from each reviewer below. During our discussion, the reviewers agreed that this was potentially an important result. We felt that the manuscript would be substantially improved if your team could:i) Provide more information about the location (regions and subnetworks), scale and direction of the spatial variation that is most consistently observed across participants;

As requested, we have included a complete set of figures in Figure 2—figure supplement 1–Figure 2—figure supplement 6. These figures summarise structured, behaviourally-relevant spatial variability for all signal PFMs, providing a thresholded view of extremes from the supplementary movies. In addition, we have included Figure 2 in this revised manuscript as a summary of topographic variance.

ii) Clarify the logic behind the central claim [that spatial variation accounts for the behaviorally-relevant variation in edge-strength] and show that the analytics are not biased to translate non-spatial sources of variability into spatial sources of variability;

We believe that the key finding is that cross-subject variability in network matrices (based on group-level parcellations) largely reflects variability in spatial topography of functional brain regions, and that this is directly supported by the results presented in this manuscript. As raised by one of the reviewers, there is indeed a possibility that PROFUMO represents connectivity information in the spatial maps. Therefore, we implement multiple additional analyses in order to demonstrate that such method-specific effects are highly unlikely to be driving our findings. Firstly, we use thresholded and binarised maps in the simulation framework, to remove the most likely way in which edge information may be represented spatially (i.e. by variations in the relative amplitudes of different regions). The results (Supplementary file 1) show that similar findings are obtained using binarised maps. Secondly, we perform an entirely separate analysis that avoids biases or underestimation of effects that may be introduced by PROFUMO. The finding (Figure 4) that unique spatial variation in task basis maps continues to be associated with measures of behaviour, while unique network matrix information is not, provides (in our opinion) unbiased and strong support for the core finding in this work.

iii) Justify the use of low-dimensional decompositions in some analyses and show whether fine-grained parcellations could ameliorate (or exacerbate) the misattribution effects that are reported;

In this revised manuscript, we have justified the dimensionality for the task basis maps in Figure 4. The dimensionality for this analysis (d=15) was primarily chosen because more than 99% of the total variance (at the group level, across all contrasts) is explained at this dimensionality. In addition, please note that our first set of simulations were also performed on the HCP_MMP1.0 (Glasser, 2016) parcellation, which has a high dimensionality of 360 parcels. Based on the wide range of dimensionalities and approaches tested in our simulations (Supplementary file 1), and the entirely independent validation of our core conclusions using an unbiased approach (based on task data), we believe that our conclusions are robust across decomposition methods and dimensionalities.

iv) Provide more information about which specific behavioral metrics covary with the spatial/morphological variation.

In this revised manuscript we have included a visual summary of the strongest behavioural measures associated with the CCA mode of population covariation that was consistently identified from a variety of fMRI-derived measures (Figure 1). Additionally, in Figure 1—figure supplement 1, we have now included a direct quantitative comparison of the observed behavioural CCA result across all instances of CCA included in Figure 1.

We also recommend trying to make the manuscript (especially the Introduction and Results) more readable for those who are familiar with fMRI and functional connectivity methods, but who are not experts in the individual differences sub-field and who have not yet read the Materials and methods in detail.

We thank the reviewers and editors for their feedback and suggestions to improve the manuscript. We have followed these suggestions by introducing the CCA approach early on in the Introduction section, and by including further descriptions of key CCA variables at the start of the Results section.

Reviewer #1:This manuscript presents a potentially interesting study on the spatial arrangement of functional regions and how they interact, and the relationship of this to non-imaging measures. Overall I think I agree with the premise of the paper (anatomy and function are not independent nor completely separable) and find the result of interest to the brain-behaviour community. However, the paper is really hard to read/follow and its unclear to what extent the authors have proven their point.

As suggested by the reviewers and editors, we have revised the writing to clarify the Introduction and Results (in particular with relevance to the use of the CCA method to test associations with individual differences in behaviour). While we fully appreciate that this work is inherently relatively complex, we hope that these changes help the reader to digest our findings. All changes are highlighted in blue in the revised manuscript (see Related Manuscript File).

The difficulty in following this also makes it very difficult to evaluate the significance of the findings. For example in the first simulation topographic information is added to the HCP data and then it is shown that topographic information accounts for some of the variance. It's not clear how this wouldn't be the case since that is what was added. While the simulation may indeed show that topology accounts for a significant fraction of the variance, this doesn't actually show that this is what is happening in real data. In general, the idea that spatial topology matters, while likely true, is poorly supported by an ad-hoc combination of ICA, parcellation, and PFMs.

We should clarify that no information was *added* to the data in our first set of simulations. The topographic information is one of the three elements that is extracted from the original data using the PROFUMO decomposition framework (in addition to time courses and amplitudes). In our simulations we hold some of these elements fixed to *remove* cross-subject variability driven by one or more of these three elements. Our findings show that when both time courses and amplitudes are fixed such that only the cross-subject spatial variability from the original topographic information is present in the simulated data, the network matrices obtained from this data are highly similar to the original connectivity matrices.

Besides the intractable writing there are at least two major concerns:1) The idea that individualized parcellations can reveal information has been shown previously. Comparing networks derived from group-parcellations with individualized spatial maps is not really a fair comparison. While it makes a nice argument for performing individualized functional parcellations the comparison as set up is not that interesting nor informative. To show that the correlation matrices (networks) do not add any unique information, one would need to show that the differences in parcellations (ICA-component definitions) across subjects can predict behavior better than, or comparable to, networks derived from the individualized functional maps. This is not what was done however.

We thank the reviewer for their comment. We would argue that our findings do show that network matrices do not add unique cross-subject information over and above spatial variability, and in this revision we have tried to make the text clearer on this point. In particular, the analysis we performed in Figure 4 does exactly as suggested: subject-specific spatial maps are derived from the individual-subject task responses (which we argue is something of a gold standard in this context), and we compare the unique information contained in the maps themselves (“Subject-task-based rfMRI spatial maps”) with the network matrices derived from these individualised functional maps (“subject-task-based rfMRI full/partial netmat”). The red bars in this figure show CCA results against behaviour when only the unique variance is used in the analysis (i.e., after removing the variance that is shared between the maps and networks). The results show that the unique information in spatial maps is highly significant, but the unique information in networks obtained from individualised functional maps are no longer significant. In other words, differences in spatial maps predict behaviour better than differences in network matrices.

2) It also appears to be the case that the authors derive the truest subject-specific map (Figure 4) from a group-level ICA. This is mentioned briefly in the Materials and methods section and is necessary to preserve the correspondences in ICA components across subjects. It appears that to maintain correspondence ICA was first performed at the group level and then brought to the individual level. It is not clear that this is a fair approach.

A traditional fMRI task analysis pipeline first performs a GLM at each voxel/grayordinate in the brain, which generates subject-specific contrast maps (note that these, purely temporal, analyses are entirely unaffected by any cross-subject variability in spatial topography, and are often carried out in subjects’ native-space data). These subject-specific task contrast maps are then entered into a standard-space cross-subject GLM to obtain group average contrast maps (which are likely to be affected by topographic cross-subject variability).

In our approach, we assume that the gross activation structure during the tasks is captured by the group average contrast maps. We believe this is a reasonable assumption given the strong significance and replicability of the group average contrast maps. As the reviewer highlights, the advantage of this task-based approach is that group and subject task contrast map are equivalent by construction, meaning that any combination (described by the ICA weights) that is valid at the group-level is equally valid at the subject-level (with the advantage of capturing subject-specific spatial features at the subject level). In the absence of ground-truth knowledge of spatial topography, using the task contrasts as subject-specific functional localisers is, to our knowledge, the most unbiased method for obtaining subject-specific maps. The reasoning behind this approach has been clarified in the revised manuscript:

“The advantage of using basis maps derived from task data is that the tasks essentially act as functional localisers that allow for the precise localisation of task-related functional regions within an individual; results at a single-subject level are not influenced in any way, including spatially, by the group results, as they are derived via the standard task-paradigm analysis (i.e. which relies solely on temporal information, and is not influenced by the group-level maps). The equivalence between group- and subject-level contrasts (i.e. the inherent assumption in any group-level analysis, namely that the group “2BK-0BK” contrast map directly relates to any subject-level “2BK-0BK” contrast) means that any combination of group-level contrasts is equally valid as a combination at the subject-level, but with the advantage that the resulting subject maps will be faithful to the precise location of functional regions that the subject-specific contrast maps capture.”

Finally, the reviewer notes that the “Group-task-based rfMRI spatial maps” give the strongest full-CCA result with behaviour. However, the partial-CCA corroborates our simulation analyses: the dual-regression derived spatial maps appear to be conflating spatial and temporal variability, given that regressing out the network matrices strongly reduces the amount of behavioural information. The subject-derived task maps contain by far the most *unique* spatial information. This result is highlighted in the revised manuscript:

“While the full CCA result is marginally stronger for group-task-based rfMRI spatial maps compared with subject-task-based rfMRI spatial maps, these group derived maps do not contain a large amount of unique spatial information (as shown by the reduced partial CCA result).”

The CCA's in Figure 1 are interesting but somewhat unclear. Independent of the concerns with the simulation expressed above, the interpretation of Figure 1 seems to be that the functional data is confounded by the topographic information but this sort of summary does not show that. I think the authors are implying that the high CCA for topography means that this is the source of the functional CCA but these do not have to be from the same source. The summary data needs to be broken down further to prove that argument.

The main interpretation we draw from Figure 1 is that the different types of measures extracted from rfMRI data (including network matrices and topographic information) all contain highly similar information. This finding subsequently led us to perform the simulation analysis and further steps presented in the rest of the manuscript. The conclusion that network matrices (derived from group maps) are confounded by spatial topography is based on the full set of results presented in this work (in particular from Table 1 and Figure 4). We have clarified this:

“Overall, these findings reveal that a large variety of fMRI measures have similarly strong associations with behaviour.”

The only functional (nonICA atlas used) is the Yeo atlas, which was somehow extended from 17 regions up to 109, but this is really a low resolution atlas. It has been shown by many groups that functional atlases of this order (such as AAL) are very poor and there are numerous publicaly available atlases with many more nodes such as the Shen atlas (Shen, 2013, which should be mentioned in the Introduction) 268 nodes, the Glasser atlas (360 nodes), which is discussed but not apparently used. This is particularly important given the comment in subsection “Cross-subject information in fMRI-derived measures” that the dimensionality of the decomposition may influence the results.While ICA 200, is high relative to the low 25 ICA it is still not high relative to many functional parcellation schemes.

The Glasser atlas (360 nodes) was used to assess the consistency of the findings across different dimensionalities. This atlas was referred to as HCP_MMP1.0 in Supplementary file 1. We have clarified this by merging the supplementary text into the main manuscript:

“Direct comparison between the results in Figure 1 (Supplementary file 1) and the HCP_MMP1.0 parcellation (e.g. the 360-region ‘Glasser parcellation’ (Glasser et al., 2016)) and against associated fractional surface area (in native space as a ratio to total surface area, for each of the 360 parcels in the HCP_MMP1.0 parcellation) is challenging due to the large difference in the number of subjects (n=819 for Figure 1=441 for HCP_MMP1.0). Therefore, we have included an analysis on all PFM metrics in a reduced number of subjects (the same n=441 subjects) in order to facilitate direct comparison between these two recent parcellation approaches that both aim to achieve accurate detection of subject-specific spatial boundaries (Supplementary file 1). These results show that spatial features from a variety of sources (surface area, multimodal parcellation and PFMs) are strongly associated with measures of behaviour and lifestyle. Also note that network matrices obtained by the HCP_MMP1.0 parcellation are more predictive of behaviour than are PFM network matrices.”

We thank the reviewer for pointing out the Shen atlas, which we have included in the Introduction:

“Other approaches are available to obtain group and subject parcellations in one step, for example using a groupwise normalised cut spectral clustering approach (Shen, Tokoglu, Papademetris, and Constable, 2013).”

As described in the Materials and methods, the Yeo atlas was subdivided into 109 nodes by separating each of the 17 networks into contiguous nodes with a surface area of at least 20 mm^2^.

Reviewer #2:This manuscript directly assesses the relationship between underlying spatial topography and functional connectivity. In general, from my knowledge of the literature, this is the first time that such a direct comparison has been performed in the context of brain-behavior comparisons. Overall, I thought the manuscript was well-written and the analyses well-defined and well-executed, and I believe that this topic (and these findings) are of particular relevance to the wider neuroimaging community, especially given the widespread use of this dataset from the Human Connectome Project. Some notable remarks include the use of high-quality publicly available data from a large cohort, and the reproducibility of the findings across parcellations (and thresholding/binarization). Please find below comments for the authors to consider.1) What was the justification for the use of a dimensionality of 15 for the creation of the ICA basis maps? It would be good to add this to the Materials and methods section "Unique contribution of topography versus coupling" or within the results (say in subsection “Unique contribution of topography versus coupling”).

This dimensionality was adopted based on the proportion of variance explained in the PCA data-reduction step. As can be seen in Author response image 1 dimensionality of 15 explains 99% of the total variance, and was therefore chosen for the further analyses. While there are 86 task contrasts, we only need relatively few components for two reasons: a large number of the contrasts are simply negations of one another, and the tasks themselves fall under only 7 broad classes. This information has been added in the main manuscript:

“The ICA dimensionality was determined based on the proportion variance explained in the PCA data reduction step (99.0% for d=15).”

2) How consistent were the loadings of the behavior CCA outputs from each of the comparisons with brain (i.e., Figure 1 in this manuscript)? Is it possible to quantify these relationships across the comparisons in this work? It would be good to evaluate this empirically, with reference to the CCA loadings in Smith et al., 2015 as reference in this manuscript.

We thank the reviewer for this point and agree that the similarity of the subject loadings of CCA results is of interest to this work. In this revised manuscript, we include a quantitative comparison obtained by correlating the subject weights across different instances of the CCA. The results show very strong similarity across all the results, and in comparison to the loadings for the ICA200 partial network matrices (i.e. the Smith et al., 2015 finding in the full 819 sample). This information is added in the supplementary material as Figure 1—figure supplement 1 and described in the text:

“Correlating the behavioural subject weights (V) across the different CCA instances in Figure 1 shows that a similar behavioural mode is obtained from the independent instances of CCA (particularly for those CCAs that have a high r_U-V_ and low P_U-V_; Figure 1—figure supplement 1).”

In addition to this quantitative analysis, the manuscript would benefit from more of a discussion (and potentially a supplemental figure) about the resultant behavior axes derived from the CCA (i.e., similar to Figure 1 from Smith et al., 2015).

As suggested by multiple reviewers, we have included a figure and accompanying description of the main (average) behavioural axes obtained across the CCAs in the main manuscript Figure 1 and in the text:

“Correlating the behavioural subject weights (V) across the different CCA instances in Figure 1 shows that a similar behavioural mode is obtained from the independent instances of CCA (particularly for those CCAs that have a high r_U-V_ and low P_U-V_; Figure 1—figure supplement 1). Mapping these subject weights onto behaviour through correlation reveals consistent positive associations with, for example, fluid intelligence, life satisfaction, and delayed discounting, and consistent negative correlations with use of tobacco, alcohol and cannabis. All behavioural correlations with mean correlation r>|0.25| (chosen for visualisation purposes) are shown in Figure 1.”

3) Given the focus/topic of this manuscript, and as briefly touched on in the discussion (and Figure 1—figure supplement 1) about the relationship between fractional surface area and subject CCA weights, it would be of great interest to the audience of this manuscript to quantify and report the relationship between basic morphological features (for example, cortical thickness, surface area, volume, myelin maps, and/or curvature) and the spatial rfMRI maps/networks. Clearly spatial variability explains a great deal of variance in rfMRI-derived network matrices, but it would be good to compare these maps to classical morphological metrics, which are derivable from these data.

We agree with the reviewer that the relationship with morphological features is of great interest following our results, and warrants detailed further investigation. We performed multiple tests, including running CCAs on displacement maps (from MSMSulc to MSMAll) in X, Y, and Z directions, Jacobian maps, warp fields (from native space to MSMAll), and parcellated fractional surface area. Significant CCA results were obtained from all of these tests. While an even more comprehensive investigation is beyond the scope of the work presented here, our fractional surface area results for parcels (Figure 2—figure supplement 7) suggest that anatomical variability in the cortical extent of a number of higher level sensory and cognitive brain regions may contribute to the spatial information obtained from functional information.

4) I would highly recommend the authors include a section (in the Materials and methods) about the data quality assessment and quality control that was performed, as well as elaborate on the preprocessing procedures. This would be of particular interest to those that would like to try to reproduce these findings or replicate them in an independent dataset.

We agree with the reviewer that quality assessment is an increasingly important stage in projects such as the HCP. A detailed overview of quality control for the HCP has been published previously by Marcus et al., 2013. For example, all structural scans are reviewed manually for image quality and artefacts. T1 weighted and T2 weighted images that are rated below 3 (on a 1-4 scale) were replaced by a new scan (typically obtained during the day 2 visit). We have added this reference and some additional relevant information in the main manuscript:

“ICA-FIX achieves 99% sensitivity and 99% specificity on HCP data when compared to manual classification by trained raters (Smith, Beckmann, et al., 2013). Only subjects with the full 4800 resting state timepoints (4 scans of 1200 TRs each) were included for the analyses performed in this work. A detailed overview of quality assessment in the Human Connectome Project was previously published (Marcus et al., 2013).”

Reviewer #3:The study by Bijsterbosch and colleagues addresses the important question of what aspect of functional connectivity varies across individuals. Prior work has largely investigated variance in the strength or amplitude of functional connectivity, however, the current findings demonstrate that much of that variance is better explained by differences in spatial topography. The analyses are conducted on data made available by the Human Connectome Project, and include validation using several network decomposition and parcellation approaches.The topic of this study is of high relevance to the field, and my primary recommendation to the authors would be to provide more extensive illustration of the topographic variance in the main manuscript. Specifically, Figure 2 is an excellent example of the type of spatial variance underlying individual differences in functional connectivity measures. Further figures depicting the spectrum of spatial variance (based on the maps shown in the supplementary videos, for example) would further help the reader understand the nature of these patterns of interindividual difference. In addition, an additional figure that summarizes the topographic variance across the cortex, along the lines of Figure 1—figure supplement 1, would be a helpful addition to the current manuscript.

We thank the reviewer for their positive assessment of our work. As suggested, we have included detailed figures of the spatial variance for all signal maps in Figure 2—figure supplement 1–Figure 2—figure supplement 6. Similarly to Figure 2, these are thresholded stills representing the extreme ends of the CCA continuum for each mode and provide a qualitative summary of the spatial variance. Additionally, we have included Figure 2—figure supplement 7 to the main manuscript (now Figure 2) as a summary description of spatial variability across all PFM modes.

Reviewer #4:Main IssuesIt is important to show that the PROFUMO decomposition is not biased to ascribe variance to the spatial mode when that variance arises elsewhere (e.g. from changes in inter-regional correlation). Please include a simulation which explicitly tests this idea (or explicitly describe for the reader the simulations from the prior PROFUMO papers that have demonstrated this fact). It is particularly important to demonstrate this for the case of overlapping generator ROIs. For, example, one could run a simulation on a 40 by 40 voxel grid, with Gaussian sources A, B, C, D located with centers as (10,10), (10,30), (30,10) and (30,30) respectively. The sources are expressed in space as Gaussians with s.d.s of (say) 10 units so that the signals from different sources do overlap. Now, we suppose that the four source signals are generated as a coupled linear system with some coupling matrix. We then ask whether changing the coupling matrix (without changing the spatial profile of the sources) leads PROFUMO to change its inferred spatial modes.For example, in condition 1, the B source is correlated with the C source, and the A source is correlated with the D source. In condition 2, the B source is additionally correlated with the A source. Are the _shapes_ of the sources recovered by Profumo different in Condition 1 and Condition 2? This is important to show, because otherwise the spatial variation inferred by PROFUMO may actually arise from changes in the true coupling matrix. Are the spatial maps the same even under conditions where the true coupling strengths fluctuate around their means over time?

We thank the reviewer for their suggestion, and we agree that it is important to ensure that our results are not driven by effects that are specific to the PROFUMO decomposition. Indeed, this was the reason for deciding to perform an entirely separate analysis based on the task contrast maps (‘Unique contribution of topography versus coupling’). The group-ICA decomposition in that analysis is blind to subject variability in both network matrices and spatial maps, to ensure that neither of these aspects drive the decomposition. Additionally, the subject-specific contrast maps are not at all dependent on group maps (because group maps are not used to estimate them), which also avoids any underestimation of spatial variability, for functional areas that are well-characterised by the tasks used. While we agree that different methods may ascribe variance differentially (as we discuss in the Results section), we hope the replication of our core finding using an entirely independent (as arguably unbiased) approach addresses the worries raised by the reviewer.

In addition, we specifically address the possibility of PFM maps reflecting network information spatially (through variability in the amplitude of subregions) by feeding thresholded and binarised maps into the simulation framework. These maps contain identical values for all suprathreshold grayordinates, thereby removing any within-network connectivity information. The results (Supplementary file 1) remain similar, showing that variability in network matrices can be explained, to a large degree, by spatial variability in binarised maps.

With regard to the simulations specifically, this was something we sought to address with the detailed simulation work in the original PFMs manuscript (Harrison et al., 2015). There, we examined the behaviour of the PFM decomposition on simulated data that included cross-subject variability in the spatial locations and relative strengths of functional regions, as well as temporal variability in both the haemodynamics and functional coupling (crucially, for the question raised by the reviewer here) between modes. What we demonstrated was that PROFUMO was able to more accurately infer the spatial maps and network matrices than ICA-based approaches, despite significant variability in both domains simultaneously. This information was added to the revised manuscript:

“Previous simulation results have shown that PROFUMO is able to accurately estimate spatial maps and network matrices in the presence of cross-subject variability in spatial topography, relative strength of subregions, and between-mode connectivity (Harrison et al., 2015).”

The prediction of behavioral properties using fMRI is a central aspect of this manuscript, and yet there is almost no description of what aspects of human behaviors are being predicted. Which aspects of task performance are being well-captured? In subsection “Cross-subject information in fMRI-derived measures” there is reference to the in "positive-negative mode of covariation" but there is no description of what this means concretely.

We agree with all reviewers that a more detailed description of the behavioural mode of covariation captured in our results may be of interest to some readers (though the specific behaviours are not too central to the main points of the work here). In the revised manuscript this is included in Figure 1 and in the text:

“Mapping these subject weights onto behaviour through correlation reveals consistent positive associations with, for example, fluid intelligence, life satisfaction, and delayed discounting, and consistent negative correlations with use of tobacco, alcohol and cannabis. All behavioural correlations with mean correlation r>|0.25| (chosen for visualisation purposes) are shown in Figure 1.”

Some of the writing in the Results section seems to assume that the readers have already read the Materials and methods, and yet the Materials and methods appear after the results. I suggest providing the readers with a bit more methodological context in the following subsections, to help them understand what was done:In the Introduction section, when introducing the PROFUMO method, please elaborate a little more on what it does and how it works;

We thank the reviewer for these specific suggestions to improve our manuscript. As suggested, we have included further detail on the PROFUMO method in the revised manuscript:

“Another approach that aims to achieve a more accurate subject-specific description of this spatial variability is PROFUMO, which simultaneously estimates subject and group probabilistic functional mode (PFM) maps and network matrices (instead of separate parcellation and mapping steps). Specifically, PROFUMO is a matrix factorisation model that decomposes data into estimates of subject-specific spatial maps, time courses, and amplitudes using a variational Bayesian approach with both spatial and temporal priors that seek to optimise for both spatial map sparsity and temporal dynamics consistent with haemodynamically-regularised neural activity (Harrison et al., 2015). PROFUMO adopts a hierarchical approach by iteratively optimising subject and group estimates (instead of first estimating group components using group ICA and separately mapping these onto subjects using dual regression), and is therefore expected to more accurately capture subject-specific spatial variability than does dual regression.”

Relatedly, when comparing PROFUMO with dual regression approach, please provide readers with a brief description of the dual regression method, so that they can understand why it may not succeed in accurately recovering subject-specific networks;

In response to this comment, we have included a description of dual regression early on in the Introduction section of the revised manuscript:

“The first stage of a dual regression approach involves multiple spatial regression of group ICA maps into each preprocessed individual dataset to obtain subject-specific timeseries; the second stage is a multiple temporal regression of these stage 1 timeseries into the same preprocessed dataset to obtain subject-specific spatial maps. Note, dual regression is, to some extent, expected to underestimate subject-specific spatial variability because it involves post-hoc regressions of a group-level set of spatial maps, which are unlikely to be an accurate model for the data of individual subjects.”

In addition, we included a brief description of dual regression in the manuscript where relevant.

Introduction section: "simulations that manipulate aspects of the data" – please be more precise about what is manipulated in the simulations;

In the revised manuscript, we have included addition detail to describe the simulations more accurately, for example:

“We then use simulations that manipulate aspects of the data such that, for example, only cross-subject spatial variability is present in the data (i.e., by fixing edge strength to be the group average for each individual) to investigate whether these differences reflect meaningful cross-subject information and drive edge estimates for several common FC approaches.”

Figure 1 – this Figure could be more effective if there was a graphical depiction of how the predictions are made from each of these fMRI-derived measures; as it is now, the reader arrives at this point in the manuscript with little idea of what is meant by any of the row-labels in this Figure;

In response to comments from all reviewers and editors, we have included more detail about the CCA approach in the Introduction and Results sections. In addition, we have included a description of the behavioural CCA mode in Figure 1. We hope that these combined changes improve the effectiveness of this figure.

Results section, paragraph three – once again the reader will have difficulty understanding what is being described because the relevant methods have not yet been laid;

Based on the suggestions from all reviewers and editors, we have included additional information at the start of the section on CCA in the Results:

“CCA works by finding a linear combination of behavioural measures (V) that is maximally correlated with a linear combination of rfMRI-derived measures (U). CCA scores for each subject are obtained for the behavioural and fMRI-derived measures (V and U), which represent the subject’s position along the population continuum for the latent CCA variable(s). The key result of a CCA analysis for each mode of covariation is the correlation between U and V, denoted r_UV_, which describes the strength of the multivariate brain-behaviour relationship. Given that CCA explicitly optimises r_UV_, it is essential to perform permutation testing in order to test the significance of the CCA result. To determine which behavioural measures contribute strongly to the CCA result, V is subsequently regressed into original non-imaging variables (Figure 1; although interpretation of these results is complicated by behaviour-behaviour correlations). Additionally, U is used to visualise variation at both the population extremes (see Figure 2 and Figure 2—figure supplement 1–Figure 2—figure supplement 6), and across the full population continuum (Supplementary video files).”

Subsection “Spatiotemporal simulations demonstrating potential sources of variability in edges”; "thereby allowing each aspect to be fixed to the group average prior to generating simulated data using the outer product" – at this point in the manuscripts readers do not have any understanding why an outer product is relevant, or which matrix equation is being referenced;

We thank the reviewer for their comment, which is hopefully helped by the added information on PROFUMO in the Introduction:

“Specifically, PROFUMO is a matrix factorisation model that decomposes data into estimates of subject-specific spatial maps, time courses, and amplitudes using a variational Bayesian approach with both spatial and temporal priors that seek to optimise for both spatial map sparsity and temporal dynamics consistent with haemodynamically-regularised neural activity (Harrison et al., 2015). PROFUMO adopts a hierarchical approach by iteratively optimising subject and group estimates (instead of first estimating group components using group ICA and separately mapping these onto subjects using dual regression), and is therefore expected to more accurately capture subject-specific spatial variability than does dual regression.”

In addition, we have included an explicit reference to the relevant equation and section in the Material and Methods:

“Note, we used PFMs in order to generate simulated data because the PROFUMO model separately estimates spatial maps, network matrices and amplitudes, thereby allowing each aspect to be fixed to the group average prior to generating simulated data using the outer product (as described in detail in equation [1], and in the section on ‘Creating simulated data’ in the Material and Methods).”

The authors have quite convincingly shown that spatial variation in fMRI sources across subjects is an important challenge and widespread confound. Please provide some elaboration on your comments in the penultimate paragraph of the Discussion and provide the reader with some specific suggestions on how this problem might be overcome, and in what contexts it is more or less perilous. Is PROFUMO the answer? Is functional co-registration using localizers the answer? Could brain-wide functional hyper-alignment / shared response modeling provide another solution to the spatial variation problem? e.g. Chen et al., 2015.

It is clear from the recent work in all these areas that the problem is of a high priority to the field and that a lot of effort is going in to addressing the problem of spatial variability and obtained accurate network matrix measures. While many of the recent methods (such as PROFUMO, MSM alignment, hyper-alignment, etc) offer valuable improvements/insights, it is not currently possible to point the reader at a single comprehensive solution. Before making more specific suggestions, we think that more work is required to further develop, test and compare these methods. We have included the hyperalignment option, including the suggested reference, in the revised manuscript:

“It is encouraging that significant efforts have recently gone into the methods for more accurately estimating the spatial location of functional parcels in individual subjects in recent years (Chong et al., 2017; Glasser et al., 2016; Gordon, Laumann, Adeyemo, Huckins, et al., 2016; Hacker et al., 2013; Harrison et al., 2015; Varoquaux, Gramfort, Pedregosa, Michel, & Thirion, 2011; Wang et al., 2015), and into advanced hyperalignment approaches (Chen et al., 2015; Guntupalli et al., 2016; Guntupalli & Haxby, 2017).